



# Compound events in Germany in 2018: drivers and case studies

Elena Xoplaki[1,2], Florian Ellsäßer[2,a], Jens Grieger[3], Katrin M. Nissen[3], Joaquim G. Pinto[4], Markus Augenstein[4], Ting-Chen Chen[4], Hendrik Feldmann[4], Petra Friederichs[5], Daniel Gliksman[6,7], Laura Goulier[8], Karsten Haustein[9,b], Jens Heinke[10], Lisa Jach[11], Florian Knutzen[9], Stefan Kollet[8], Jürg Luterbacher[12], Niklas Luther[2], Susanna Mohr[4,13], Christoph Mudersbach[14], Christoph Müller[10], Efi Rousi[15], Felix Simon[14], Laura Suarez-Gutierrez[16,c,d], Svenja Szemkus[5], Sara M. Vallejo-Bernal[17,18], Odysseas Vlachopoulos[2], Frederik Wolf[17]

[1] Department of Geography, Climatology, Climate Dynamics and Climate Change, Justus Liebig University Giessen, Giessen, Germany
[2] Centre of International Development and Environmental Research, Justus Liebig University Giessen, Giessen, Germany
[3] Institute of Meteorology, Free University of Berlin, Berlin, Germany
[4] Institute of Meteorology and Climate Research (IMK-TRO), Karlsruhe Institute of Technology (KIT), Karlsruhe, Germany
[5] Institute of Geosciences, University of Bonn, Bonn, Germany
[6] Institute of Hydrology and Meteorology, Faculty of Environmental Sciences, Dresden University of Technology, Tharandt, Germany
[7] Institute of Geography, Dresden University of Technology, Dresden, Germany
[8] Institute for Bio- and Geosciences, Research Centre Jülich, Jülich, Germany
[9] Climate Service Center Germany (GERICS), Helmholtz-Zentrum Hereon, Hamburg, Germany
[10] Potsdam Institute for Climate Impact Research (PIK), Member of the Leibniz Association, Potsdam, Germany
[11] Institute of Physics and Meteorology, University of Hohenheim, Stuttgart, Germany
[12] Science and Innovation Department, World Meteorological Organization (WMO), 7bis Avenue de la Paix, 1211 Geneva, Switzerland
[13] Center for Disaster Management and Risk Reduction Technology (CEDIM), Karlsruhe Institute of Technology, Karlsruhe, Germany
[14] Department of Hydraulic Engineering and Hydromechanics, Civil and Environmental Engineering, Bochum University of Applied Sciences, Bochum, Germany
[15] Potsdam Institute for Climate Impact Research (PIK), Member of the Leibniz Association, Potsdam, Germany
[16] Max-Planck-Institut für Meteorologie, Hamburg, Germany
[17] Research Department IV - Complexity Science, Potsdam Institute for Climate Impact Research (PIK), Member of the Leibniz Association, Potsdam, Germany
[18] Institute of Geosciences, University of Potsdam, Potsdam, Germany
[a] now at: Department of Natural Resources, ITC - Faculty of Geoinformation Science and Earth Observation, University of Twente, The Netherlands
[b] now at: Institute for Meteorology, University of Leipzig, Leipzig, Germany
[c] now at: Institute for Atmospheric and Climate Science, ETH Zurich, Zurich, Switzerland
[d] now at: Institut Pierre-Simon Laplace, CNRS, Paris, France

*Correspondence to*: Elena Xoplaki (elena.xoplaki@geogr.uni-giessen.de)

**Abstract.** The European continent is regularly affected by a wide range of extreme events and natural hazards including heatwaves, extreme precipitation, droughts, cold spells, windstorms, and storm surges. Many of these events do not occur as single extreme events, but rather show a multivariate character, the so-called compound events. Within the scope of the interdisciplinary project climXtreme (https://climxtreme.net/), we investigate the interplay of extreme weather events, their



characteristics and changes, intensity, frequency and uncertainties in the past, present and future and associated impacts on various socio-economic sectors in Germany and Central Europe. This contribution presents several case studies with special emphasis on the calendar year of 2018, which is of particular interest given the exceptional sequence of different compound

events across large parts of Europe, with devastating impacts on human lives, ecosystems and infrastructure. We provide new evidence on drivers of spatially and temporally compound events (heat and drought; heavy precipitation in combination with extreme winds) with adverse impacts on ecosystems and society using large-scale atmospheric patterns. We shed light on the interannual influence of droughts on surface water and the impact of water scarcity and heatwaves on agriculture and forests. We assessed projected changes in compound events at different current and future global surface temperature levels,

demonstrating the importance of better quantifying the likelihood of future extreme events for adaptation planning. Finally, we addressed research needs and future pathways, emphasising the need to define composite events primarily in terms of their impacts prior to their statistical characterisation.

## 1 Introduction

Extreme temperatures, strong extratropical low pressure systems and their associated extreme winds and heavy precipitation events can have devastating socio-economic impacts. Moreover, the combination of otherwise regular climate and weather phenomena can unfold their effects beyond the individual events (Ridder et al., 2020) and have devastating consequences (Ridder et al., 2022; Bevacqua et. al., 2017, 2021, 2023 and references therein). Thus, human and natural systems that are usually able to handle the impacts of single extreme events are challenged by the co-occurrence of two or more extremes

(compound events, CE) which severely increase the risk of loss and damage (Toreti et al., 2019a). Events with additive and multiplicative effects are of utmost importance and can result from mutually reinforcing cycles/positive feedbacks between individual events. Interrelated events, e.g., through land surface-atmosphere interactions or atmospheric moisture conditions, modify extreme events (Wang et al., 2022). The effects may develop also through atmospheric dynamics that connect features such as the 2010 Russian heatwave and the flood in Pakistan (Barriopedro et al., 2011; Lau et al., 2012; Zscheischler et al.,

2018) or through induced responses at distant areas of significant impact to the global system (Vogel et al., 2019).

The Intergovernmental Panel on Climate Change (IPCC; Seneviratne et al., 2012) defines CEs as 1) two or more extreme events occurring simultaneously or successively, 2) a combination of extreme events with underlying conditions that amplify the impact of the events, or 3) a combination of events that are not themselves extremes but lead to an extreme event or impact when combined. This definition is embedded within the IPCC risk framework under the umbrella of a combination of multiple

drivers and/or hazards that contribute to societal or environmental risks. Also embedded in this framework is the understanding that response to an imminent risk can, in its own right, serve to reduce or to increase future risks. Zscheischler et al. (2018) further defined CEs as combinations of events that are individually not necessarily extreme, or multiple drivers and/or hazards, but in combination often lead to disproportionate impacts on people and ecosystems (Seneviratne et al., 2012; Leonard et al.,



2014; Caldeira et al., 2015). To quantify the probability of CEs in today's and future climate is of great importance specifically

adaptation planning for various sectors including agriculture, fisheries, river transport, energy supply, tourism, etc. (Zscheischler and Fischer, 2020). Recently, Zscheischler et al. (2020) extended the definition and classified CEs into 1) preconditioned events, where a weather-driven or climate-driven precondition aggravates the impacts of a climatic impact-driver; 2) multivariate events, where multiple drivers and/or climatic impact-drivers lead to an impact; 3) temporally compounding events, where a succession of hazards leads to an impact; and 4) spatially compounding events, where hazards

in multiple connected locations cause an aggregated impact. Drivers include processes, variables, and phenomena in the climate and weather domain that may span over multiple spatial and temporal scales (Zscheischler et al., 2020). Current research on weather and climate impacts, risk and damage underestimates the influence of CEs (Ridder et al., 2021) and thus it is essential to adapt models, processes and studies for research to incorporate compound weather and climate extremes in order to determine uncertainties, impacts and risks. Further, anthropogenic climate change is expected to influence the

frequency and intensity of CEs, and thus future planning for such changes requires reliable climate models, which are able to represent these hazards, their underlying drives as well as their combinations. Despite this importance, studies evaluating climate model representation of CEs are still rare (Aalbers et al., 2013; Bevacqua et al., 2023; Manning et al., 2023). Further, the impact of climate change to dynamic changes in the atmosphere and consequently to the location and magnitude of extreme events and its compounds is less well understood and thus characterised as low confidence by the IPCC (IPCC, 2021). This

holds true also for CEs, that are naturally even more complex due to their multi-variate character, also in terms of the complexity of the atmospheric circulation state. For instance, the COVID-19 pandemic has brought dynamics of compound hazards and risk-response feedbacks to the forefront of hydrometeorological hazard response and preparedness (Simpson et al., 2021; Zaitchik et al., 2022). Compound hazards are rare, and those for instance that involve a disease such as COVID-19 have no recent precedent. The more complex CEs become, the clearer are the limitations of the conventional statistical

approaches to risk assessment (Zaichik et al., 2022).

The development of integrated research on CEs is the objective of the European COST Action DAMOCLES (http://www.damocles.compoundevents.org) that bundled research efforts in this field, and towards which several of the authors actively contribute. One of the main knowledge gaps identified concerns how the compound character of events is changing in a warming world, and will continue to change during future decades. The question on how and why extreme

weather events affecting specifically Germany and Central Europe may change in a warming climate is the major topic of the climXtreme project (https://climxtreme.net/). climXtreme is funded by the German Federal Ministry of Education and Research (BMBF) and comprises 35 universities and research institutions from all over Germany. The project's vision is to address some of the topics and challenges described above providing new evidence and answers questions whether extreme events and CEs are already occurring more frequently as a result of climate change, and how anthropogenic climate change

will alter the intensity, frequency and spatial distribution of extreme and CE events in the future. A group within climXtreme joined forces to analyse CEs of temperature and precipitation as well as of precipitation and wind. To analyse hot and dry compounds, a variety of research questions and approaches are explored: at the global scale, the precursors of spatially and





temporally CEs are analysed using large-scale atmospheric patterns and jet stream states. At the European scale, the detection and identification of events and the spatial representation of key climate variables in relation to heatwaves are investigated.

Focusing on Germany, the interannual influence of droughts on surface water is analysed and the impact of water scarcity and heatwaves on agriculture and forests is studied. Further, the CEs including precipitation and/or wind as a hazard are analysed focusing on a series of windstorms and convective storms with adverse impacts on ecosystems and society.

All case studies presented in this paper are selected from the calendar year 2018, which is of particular interest given the prolonged and persistent dry and hot conditions across large parts of Europe as well as featured storms Eleanor and David

(coined Burglind and Friederike in Germany and hereafter) in January 2018 and several weeks of thunderstorm activity in May and June. 2018 was also characterized by strong wind gusts that co-occurred with heavy snowfall during the windstorm Friederike (Vautard et al., 2019), a relatively dry spring with exceptionally high temperatures followed by an extremely dry summer with very warm mean temperatures over large areas of Europe (Munich Re, 2019; Toreti et al., 2019a; Zscheischler and Fischer, 2020). Total precipitation in central Europe was at the lowest percentiles relative to the 1976–2005 distribution;

Germany experienced a reduction of precipitation of 42% in July and of 45% in August (Toreti et al., 2019a,b). The summer in Germany was characterised by the most extreme combination of high temperatures, as one of the warmest years on record (Kaspar et al., 2023), and low precipitation since 1881 (Zscheischler and Fischer, 2020). The combination of the individual events caused tremendous adverse and detrimental impacts in larger areas of western Europe with a peak over Germany and on a variety of sectors ranging from agriculture and society (Manning et al., 2018; Toreti et al., 2019b; Zscheischler and

Fischer, 2020), forests (Bastos et al., 2020; Buras et al., 2020; Senf and Seidl, 2021), fires (Munich Re 2019; Bastos et al., 2020), soil and surface water (Liu et al., 2020; Brakkee et al., 2022; Hartick et al., 2021), marine environment (Kaiser et al., 2023), traffic disruption, power outages, property damage by e.g. falling trees, fatalities (Vautard et al., 2019) as well as human health (Matzarakis et al., 2020). The exceptional heatwave of 2018 also caused many nuclear power plants to shut down because the warm water in the rivers could not adequately cool the reactors (Vogel et al., 2019).

The paper presents first the different data used and methodologies applied for the analysis of the selected CEs during 2018 followed by analyses of each case study. The case studies are clustered in temperature-precipitation and precipitation-wind CE storylines together with an assessment of their impacts in Germany.

## 2 Data and Methods

The sub-projects of climXtreme employ different methodological approaches, tailored to the corresponding research questions

ranging from better understanding of the selected events drivers to sectoral impact assessments. This section summarizes these approaches and provides a basis for the study and analysis of the selected case studies separated in temperature-precipitation and precipitation-wind CEs. The temperature-precipitation storyline includes analysis on drivers of the hot summer 2018, detection of extreme events and spatial patterns, assessment of the impact of the 2018 European drought on soil moisture and groundwater as well as sectoral impacts on agriculture and forestry. The storyline is complemented with an assessment of



model simulations to realistically represent conditions as those of 2018 in Germany. The precipitation-wind storyline comprises the analysis of intense low pressure systems in winter 2018, their life cycle and triggering role for compound precipitation and wind events, as well as severe convective storms during the 2018 warm season.

## 2.1 Drivers of the hot summer of 2018

To better understand the drivers of the hot summer of 2018, Rousi et al. (2022) identified jet states in the zonal mean zonal wind over the Eurasian sector at different pressure levels for the summer months in ERA5 data (Hersbach et al., 2020) using Self-Organizing Maps (SOMs, see Kohonen, 2013; Rousi et al., 2015). A comparative approach with different cluster numbers, clustering algorithms and initializations of SOMs led to a robust cluster of double jet states. Increased persistence of those jet states was connected to heatwave events (defined as a period of at least 3 consecutive days of daily maximum temperature

threshold exceedance > 90th percentile, following Fischer and Schär (2010) and a spatial extent over 40.000 km$^2$ within a 4° x 4° spatial sliding window, similar to Stefanon et al. (2012) across western Europe (Rousi et al., 2022).

## 2.2 Detection of spatial patterns of extreme events

The analysis of the large-scale temperature and precipitation deficit patterns and their expression during the 2018 heatwave at the European scale is based on the cross-Tail Pairwise Dependence Matrix (cross-TPDM) and Extreme Pattern Index (EPI)

proposed by Szemkus and Friederichs (2023). The TPDM can be considered as an analogue of the covariance matrix for extremes (Cooley and Thibaud, 2019). Before calculating the cross-TPDM and EPI, the ERA5 daily 2m temperature and precipitation deficits for the summer months (June-August) of 2018 are standardised and the annual cycle is removed. Precipitation deficits are calculated as the inverse of the 90-day accumulated precipitation. A singular value decomposition is performed on the cross-TPDM, the singular vectors are analysed and the EPI is calculated from the first 10 left and right

expansion coefficients.

## 2.3 Surface water storage of the dry summer of 2018

To analyse the drought characteristics of the summer of 2018, an ensemble of simulations for the hydrological year 2018/19 is used (Hartick et al., 2020). The hydrological year 2018/19 was initialised with land surface and subsurface conditions from the end of the hydrological year 2018, and simulated using different atmospheric boundary conditions derived from each

individual year of ERA5 data as climatological time series between 1996 and 2018. The ensemble of 22 realisations of the hydrological year 2018/19 as such accounts for a large part of the atmospheric uncertainty. The analysis was performed for 20 European river basins. The 2018 drought was defined as the driest 10% of the total water storage anomalies (S) occurring in 2018 within the climatological time series. Surface water availability for the 2018/19 hydrological year was represented by surface water storage ($S_u$), categorised into dry, $S_{u,d}$, and wet, $S_{u,w}$, anomalies. To ensure that an increased probability of $S_{u,d}$

in the hydrological year 2018/19 was outside of regular climate variability, we compared the $S_{u,d}$ probability distribution of



the described hydrological year 2018/19 ensemble (Case A) with the probability distribution of $S_{u,d}$ within the climatological time series (Case B), see also the corresponding section below. Two beta distributions were generated, one for each case, by applying a prior with no information. We sampled each beta distribution 10,000 times and calculated the probability that Case A > Case B to determine the confidence that the probability of a $S_{u,d}$ after a drought is greater than the climatological variability.

In addition, we obtained the uncertainty of the confidence intervals by bootstrapping 1000 times over the climatological time series. The methodology provides a probabilistic insight into the impact of a groundwater drought on future surface water resources on an interannual time scale.

**2.4 Soil moisture of the dry summer of 2018**

In addition to the dry surface water anomaly in Central Europe, soils showed moisture deficits (Liu et al., 2020; Bastos et al.,

2020; Rousi et al., 2023). This likely caused low groundwater levels (Brauns et al., 2020), as infiltration of precipitation water is considered to be the most important groundwater source in Central Europe (Brakkee et al., 2022). ERA5 soil moisture was evaluated for the four soil layers over the period 2018-2020 and compared against the climatology averaged over 1991-2020 in order to assess the strength of the soil moisture deficit and its persistence during the consecutive drought years 2018-2020. For this analysis, time series of daily means as well as centred 92-day running means were computed for all land points of the

study area 4-16° E and 45-55° N, covering Germany and adjacent regions. The evaluation of soil moisture in the lowest soil layer also gives an indication of the groundwater reservoir as it interacts with the aquifer in the modelling system (Cerlini et al., 2021).

**2.5 Agricultural and hydrological drought of the year 2018**

Lack of sufficient soil moisture, resulting from shortage of precipitation and excess evapotranspiration leads to agricultural

drought. Lack of run-off and surface water result in hydrological drought (streamflow deficits) (Seneviratne et al., 2021). To estimate the severity of agricultural and hydrological droughts across Europe during summer 2018, we employed the nitrogen version of the vegetation, crop, and hydrology model LPJmL (Schaphoff, et al., 2018; von Bloh et al., 2018; Lutz et al., 2019; Herzfeld et al., 2021). The analysis is based on 69 years (1951-2019) obtained from model simulations driven with daily temperature, precipitation, and radiation data from the GSWP-W5E5 dataset (Kim, 2017; Cucchi et al., 2020; Lange et al.,

2022) at 0.5 arc-degree resolution. To assess agricultural drought, the evapotranspiration deficit calculated as the ratio of actual evapotranspiration to potential evapotranspiration (ET/PET ratio) over the growing season of maize in each year is determined and a generalised beta distribution (a three-parameter probability distribution for variables in a bounded interval) is fitted to the 69 annual values in each grid cell. An ET/PET ratio of less than 1 indicates water deficit or water stress. For the assessment of the hydrological drought, the average river discharge (Dis) during the summer months (June, July, and August) of each year

is determined and a generalized gamma distribution (a three-parameter probability distribution for non-negative variables) is fitted to the 69 annual values in each grid cell. Using the fitted distributions, the return period of the conditions in 2018 is determined. To support comparability with other drought indices such as the Standardized Precipitation Evapotranspiration



Index (SPEI), the drought severity is also calculated, which is the probability (inverse of return period) of a given year expressed as its distance from the mean (in number of standard deviations) in a standard normal distribution (McKee at al.,

1993; Vincente-Serrano et al., 2010). For example, a return period of 44 years is equivalent to the 2.28th percentile, which is -2 standard deviations away from the mean and would be assigned a drought severity of -2.

**2.6 Impact on the agricultural production of 2018**

In comparison with the past three decades, the year 2018 was identified as a year with severe winter wheat yield losses estimated using a compilation of LOESS (locally estimated scatterplot smoothing; to take into account improvement of

agricultural practises (Zampieri et al., 2017)) detrended and gap-filled yield data at county level aggregated from a variety of sources including the Regionaldatenbank Deutschland (Statistische Ämter des Bundes und der Länder, 2021) and the Statistical Offices of the federal states of Germany (Ellsäßer and Xoplaki 2022abc; Ellsäßer and Xoplaki, 2023). The resulting annual gridded yield data was evaluated using the Standardized Yield Anomaly Index (SYAI) that expresses yield anomalies in terms of standard deviation from a 30-year time series. The analysis is based on the heat stress index Heat Magnitude Index (HMD)

(Zampieri et al., 2017), the drought index SPEI (Vicente-Serrano et al., 2010) and the Combined Stress Index (CSI) (Zampieri et al., 2017) that accounts for stress compounds of heat and drought through a (ride-regression based) superimposition of HMD and SPEI, using the temperature and precipitation series from E-OBS (Cornes et al., 2018). In order to derive crop relevant results, all indices were evaluated for the most vulnerable stages of phenological crop development according to the specific region using the German Weather Service (DWD) phenological data set (Kaspar et al., 2015). A spatially explicit linear

regression between yield anomaly and stress indices was computed for time series covering the past three decades and the coefficient of determination ($R^2$) was calculated to express the proportion of yield anomaly that can be explained by heat, drought or compound stress.

**2.7 Loss and damage of compound vs. non-compound wind extreme events of the winter 2018**

In the context of the UN Framework Convention on Climate Change (UNFCCC) process, loss and damage is the harm caused

by anthropogenic (human-generated) climate change (UNFCCC, 2021; OECD, 2021 and references therein). For the quantitative assessment of the impact of CEs in terms of loss and damage, a compound wind and precipitation extreme is defined when the local 98th percentile of both variables is exceeded (Martius et al., 2016). For winter events, the calculation of the percentile is based on data from the December to February season. The co-occurrence should be on the same day or the following day for precipitation, in the same grid cell and within a radius of 50 km, respectively. The daily loss data for

residential buildings accumulated over Germany provided by the German Insurance Association (GDV) are categorised by days on which a CE occurred and days on which it did not. This results in two separate loss distributions for compound and non-compound events. The cyclone track analysis for the 2018 winter season is based on the cyclone tracking methodology of Murray and Simmonds (1991) and Pinto et al. (2005) applied to ERA5 data (Hersbach et al., 2020).



**2.8 Concurrent heavy rain and storm extremes – estimation of probability of event occurrence**

The estimation of the probability of occurrence of compound heavy rain and wind is carried out on precipitation and wind time series from DWD weather stations. Archimedean copula functions are used, in particular the Frank copula (Frank, 1979). The Frank copula is a one-parametric copula in which the copula parameter *theta* can be determined from the correlation between random variables. The approach treats the extremes of the two variables (rain and wind) separately. The annual maximum values (AMAX) are extracted from the time series and tested for statistically significant trends using the Mann-Kendall test at

a 5% significance level. If significant trends are found, the time series are homogenised using a linear regression model. The distribution parameters of the strong wind and precipitation data sets are then determined using maximum likelihood. In addition, the correlation between heavy rain and storm is calculated using the Kendall rank correlation. For each AMAX wind value, the concurrent precipitation value is selected and vice versa. To ensure the independence of precipitation events, wet episodes are separated by a dry period of at least as long as the accumulation period. Due to this restriction, not all AMAX

wind events can be paired with a precipitation episode, even when precipitation is present, e.g. for longer precipitation durations (>1day) and considering a ±2-day window, no AMAX wind and precipitation episode pairs exist.

**2.9 Rockfall events**

Rockfall is an impact that can be triggered by extreme precipitation. A logistic regression model describing the probability of rockfall in the Central European low mountain ranges was fitted to the Rupp and Damm (2020) database. The model

incorporates the compound nature of such events by including several parameters (multivariate compounding) and taking into account preconditions (temporal compounding). The predictors for the statistical model are the across-site percentile of a fissure water proxy D (precipitation minus potential evaporation determined for the last 5 days), the local percentile of daily precipitation and the binary information if a freeze-thaw cycle has occurred within the last 9 days. Details of the statistical model can be found in Nissen et al. (2022).

**2.10 Convective cluster events of the summer 2018**

Convective cluster events (CCEs) are spatially connected areas of intense lightning activity that occur simultaneously in the same geographic region. CCEs can be detected using the Spatial-Temporal Density-Based Spatial Clustering of Applications with Noise (ST-DBSCAN) algorithm (Ester et al., 1996; Birant and Kut, 2007). The data used are cloud-to-ground lightning strokes from the European Cooperation for Lightning Detection (EUCLID) network (Schultz et. al., 2016). ST-DBSCAN is

further developed and specifically adapted for the detection of spatio-temporal clustering of lightning strokes (Augenstein et al., 2023). The algorithm identifies arbitrarily shaped clusters in a set of given points, which in this case are spatio-temporally close lightning strokes. For the identification of CCEs, thresholds from sensitivity studies have been used, i.e., if at least 40 lightning strokes occur within 20 minutes and 50 km, single lightning stokes are marked belonging to a CCE. These thresholds have proven to be an "optimal" balance to distinguish between lightning clusters and noise.





**2.11 Frequency of compound events from recent to near-term future climate conditions**

The estimation of the projected changes in the frequency of CEs from recent to near-term future climate conditions is based on the 30-member CMIP6 MPI-GE (Olonscheck et al., 2023). The historical and SSP585 (Riahi et al., 2017) simulations for the periods 1975-2025 and 2025-2075 are used, representing climate conditions that for the CMIP6 MPI-GE range from about 1 to 3 °C increase in global mean surface temperature since pre-industrial times (Olonscheck et al., 2023). The projections

cover Germany and more specifically the region defined by the 4-16° E and 45-55° N latitude-longitude domain. The compound heat and drought events are defined by the cumulative precipitation from May to October and the mean daily maximum temperature in summer (June to August), spatially averaged over Germany. Extreme compound hot and dry years exceed the 20-year return levels for both precipitation deficit and maximum temperature individually, defined as the 5th and 95th percentiles, respectively, for the period 1975-2025. The compound precipitation-wind events are defined on the winter

(December to February) daily mean precipitation and daily maximum surface wind. The selection of events is based on the exceedance of the 98th percentile for the period 1975-2025 and each grid cell on the same day for both variables or the day after for precipitation only. The cumulative effect for the whole season is the sum of all daily occurrences over all winter days and each grid cell. Extreme compound wind and precipitation years exceed the 20-year return levels for precipitation and wind individually, defined as the 95th percentiles for the period 1975-2025.

**2.12 Representation of moisture availability of 2018 in model simulations**

The performance of model simulations in realistically representing drought conditions like those of 2018 and the 2018-2020 three-year drought cluster of events is assessed based on the estimated trend of the warm season (March to August) moisture availability in Germany. Drought conditions are described with the SPEI index (Vicente-Serrano et al., 2010) for observations, using ERA5 data (Hersbach et al., 2020) for the period 1979-2019, and for bilinearly interpolated (regular 0.5° lon-lat grid)

ensemble simulations of CMIP5 (Taylor et al., 2012; Aalbers et al., 2023) global circulation models and of EURO-CORDEX 0.11° (Giorgi et al., 2009) regional mutli-model ensemble for the historical (1950-2005) and the near-to-mid-term future (2006-2070) periods under RCP8.5. The linear trend of the 3-year running mean for the March to August intervals is calculated over the periods 1975-2021 for the simulations (or 1979-2019 for ERA5) and 2022-2070 in order to account for the transition from dimming to brightening regime in the 1970s (Wild, 2009, 2016).

**3 Compound events in the year 2018**

**3.1 Temperature-precipitation compound events during 2018**

The exceptionally hot and dry conditions in 2018 extended over larger areas including central and northern Europe and were associated with impacts on various economic sectors (Toreti et al., 2019a; Zscheischler and Fischer, 2020). We study this exceptional year and the series of extremes and CEs in a sequence of the large scale, their detection and spatial representation



### 3.1.1 Drivers of the hot summer of 2018

The 2018 heatwave was a spatially CE featuring concurrent heatwaves in Scandinavia and central Europe (Spensberger et al., 2020; Rousi et al., 2023). Studies have explored the potential attribution of the strong heatwave of 2018. Prior to the 2018 heatwave, a stripped high-pressure system formed over northern Europe in late June, during a combination of the positive phase of the North Atlantic Oscillation and the Rossby Wave 7 pattern (Drouard et al., 2019; Kornhuber et al., 2019). Figure 1 presents the jet stream state during the 2018 summer and the heatwave day frequency for each grid point over the Eurasian sector. During the intense European summer heatwave, a large blocking system at 500 hPa, and a double jet stream configuration is visible in the 250mb zonal wind field (Kornhuber et al., 2019; 2020; Rousi et al., 2023, see Methods section 2.1). Heatwave hot spots over Europe coincide with areas of weak winds between the polar and subtropical jets. Such large-scale atmospheric conditions are conducive to the occurrence of extreme events over Europe, in particular to heatwaves near the centre of blocking system (see Kautz et al., 2022 for a review). In particular, the hot summer of summer 2003 (western/central Europe, Luterbacher et al., 2004; Fink et al., 2004; Fischer et al., 2007; García-Herrera et al., 2010) and 2010 (heatwave over western Russia, Barriopedro et al., 2011; Di Capua et al., 2021; Rousi et al., 2022) were characterised by similar large-scale conditions.

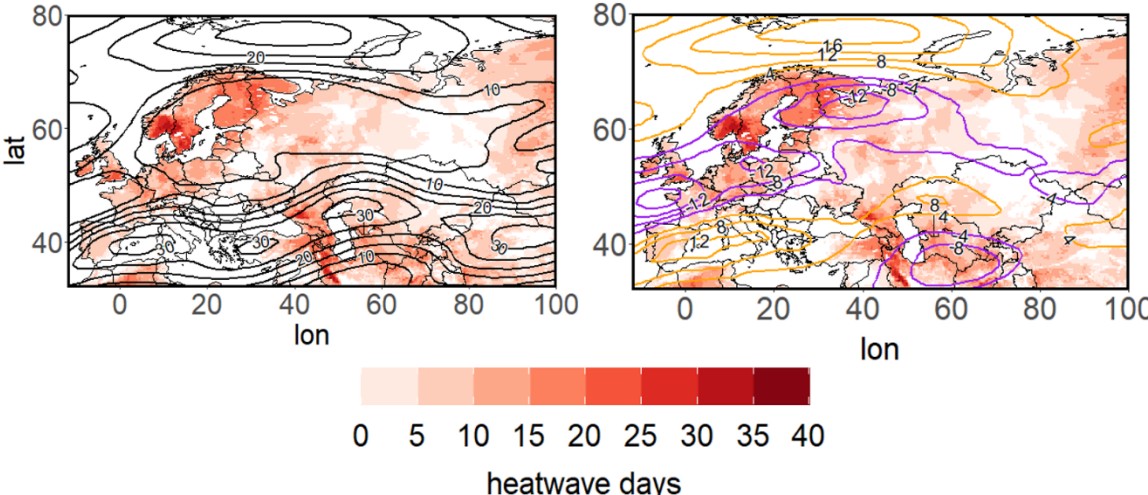

**Figure 1: Jet stream state (contour lines) and heatwave days in summer 2018 (shading). Left: zonal wind at 250 hPa (black contours from 5 m/s to 30 m/s every 5 m/s); Right: zonal wind anomalies at 250 hPa (anomalies based on 1979-2020 July climatology and plotted with contours from -16 m/s to 16 m/s every 4 m/s, negative anomalies are shown in purple contours and positive in orange) for the period 4-25 July 2018, the longest period of consecutive double jet states. All fields stem from ERA5 reanalysis data (Hersbach et al., 2020).**





### 3.1.2 Detection of spatial patterns of extreme events

During the summer 2018, large-scale temperature (T2m) and precipitation deficit (PD) patterns characterize the exceptional conditions. Figure 2 shows the analysis of these patterns and their expression during the 2018 heatwaves at the European scale
based on cross-TPDM and EPI (see Methods section 2.2).

In July-August 2018 the pronounced heatwave is accompanied by extreme $EPI^{PD}$ preceding the heatwave for several days. This heatwave is considered the most prominent event in the period under consideration (see also Liu et al., 2020). The negative anomalies in the second mode expansion coefficients (Fig. 2a,b bottom right) indicate the beginning of the heatwave, which initially affected northern Europe (i.e., Finland, Norway, and northwestern Russia). From mid of June onwards, there were
extremes in PD, particularly in Central Europe, as indicated by the third mode of the expansion coefficient (Fig. 2c,d bottom right). By the end of July 2018, the heatwave extended to Central Europe as evidenced from the abrupt change of sign in the third mode expansion coefficient (Fig. 2c,d bottom right).

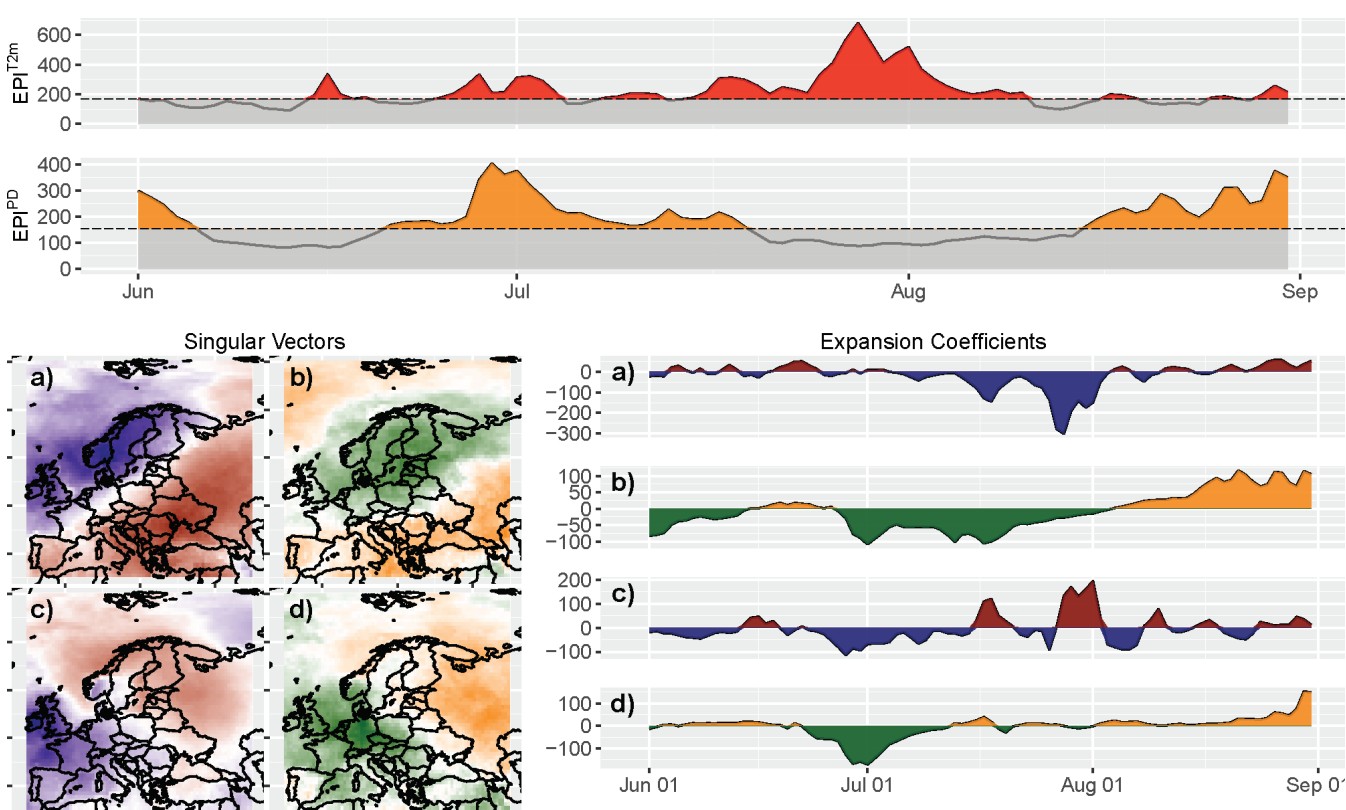

**Figure 2: Top: Extreme Pattern Index (EPI) for T2m surface temperature ($EPI^{T2m}$, red) and precipitation deficit ($EPI^{PD}$, orange)**
**for northern Hemisphere 2018 summer months. Values exceeding the 80th percentile are considered to identify extreme events in PD and T2m from EPI and are highlighted in red/orange, respectively; Bottom left: second (a, b) and third (c, d) singular vectors (CVs) associated with large-scale temperature (blue/red) and precipitation deficit (green/orange) pattern; Bottom right: second (a, b) and third (c, d) expansion coefficient for northern hemisphere 2018 summer months. Positive values are plotted in red/orange and negative values in blue/green.**



### 3.1.3 Surface water storage of the dry summer of 2018

The high temperature in 2018 was mainly due to increases in the amount of net surface radiation caused by the clear skies associated with reduced precipitation (Liu et al., 2020). Germany experienced a strong increase of net radiation of approximately +31%. Liu et al. (2020) report that land cover played a critical role in determining the occurrence and strength of soil moisture-temperature coupling, i.e. cropland/grassland depletes soil moisture more readily than forests, thereby triggering a more rapid release of sensible fluxes a major feature observed during the 2018 heatwave. During the 2018 heatwave, because of different soil moisture conditions, latent flux in Germany decreased by 12% and sensible flux significantly increased by 122% (Liu et al. 2020). Further, Bastos et al. (2020) used 11 vegetation models and showed that spring conditions promoted increased vegetation growth, which, in turn, contributed to fast soil moisture depletion, amplifying the summer drought. Figure 3 presents the groundwater memory in the summer 2019 of the ensuing hydrological year 2018/2019 for each of the 20 European river basins on the following year's summer surface water storage ($S_u$).

The ensemble simulations indicate that following the 2018 drought the conditional probability that the autumn of the hydrological year 2018/2019 (August to November 2018) is anomalously dry $p(S_{u,d})$ is 95.5% with a $100 \pm 0.0\%$ confidence with respect to the climatological variability. In the following seasons, $p(S_{u,d})$ and the associated confidence decrease due to the increasing influence of the uncertainty in the atmospheric conditions. Specifically, for winter $p(S_{u,d})$ is 81.8% with a confidence of $99.5 \pm 0.3\%$, for spring (March to May 2019) 63.6% with a confidence of $80.2 \pm 6.0\%$ and for summer (June to August 2019) of the hydrological year 2019/2020 $p(S_{u,d})$ is 68.2% with a confidence of $90.1 \pm 3.7\%$ with respect to the climatological variability. Without considering the groundwater storage memory effect, a probability of a dry surface water anomaly $p(S_{u,d})$ of ~ 50% would be expected due to the atmospheric uncertainty accounted for in the ensemble of realisations at the interannual time scale. Taking drought as a precondition for $S_u$ on this scale, the analysis shows that even one year later a $p(S_{u,d})$ of 68% is still well above 50% at a confidence level of $90 \pm 4\%$. Thus, statistically, groundwater storage takes longer than a year to fully recover from a drought influencing surface water storage, independent of the ambient atmospheric conditions (Lorenz et al., 2010; Orth and Seneviratne, 2012; Song et al., 2019). Recent evidence points to the fact, that the impact of global warming on soil moisture drought severity in west-central Europe such as the case in 2018 is increased. The drought risk is strongly enhanced by the drought intensification and increase in frequency, yielding shorter recovery time between events for nature and society (Aalbers et al., 2023).





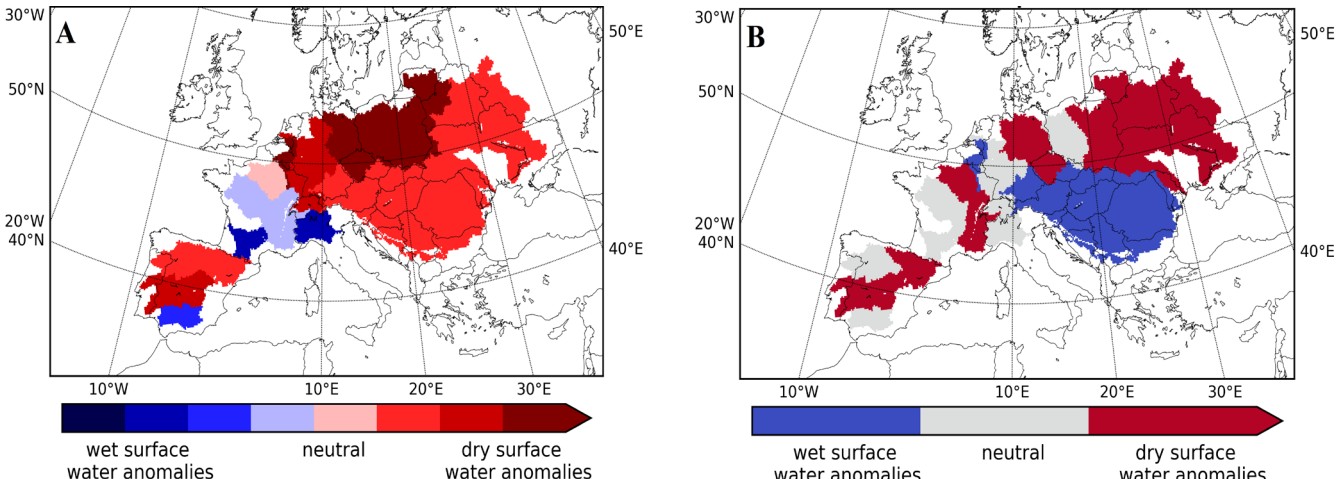

**Figure 3: Averaged impact of the yearlong 2018 drought on the following year's summer (June to August 2019) surface water storage ($S_u$) anomaly per river basin (see Methods section 2.3). A: $S_u$ anomaly in 2018 for each river basin in quartiles; B: $S_u$ anomaly in the summer (June to August 2019) after 2018 initial conditions and (ensemble) mean of 22 atmospheric conditions.**

### 3.1.4 Soil moisture of the dry summer of 2018

To address sectoral impacts, we focus on the effects of the 2018 drought on agriculture and forestry in Germany. For this purpose, the temporal evolution of soil moisture deficits at different depths from the ERA5 dataset and agricultural data from German national institutes are analysed. Focusing on the temporal evolution of soil moisture, a deficit developed during the spring and early summer of 2018 (Rousi et al., 2023), which also reached the lowest soil layer with a temporal delay of about three months, as shown in Fig. 4. The dryness in 2018 was more intense than the usual soil moisture variability in the period 1991-2020, as shown by the soil moisture dropping below the range of ±1 standard deviation of the 1991-2020 mean soil moisture (shaded area). While the moisture in the three upper soil layers mostly recovered during the following winter 2018/19, the moisture did not percolate down to the lowest soil layer, which remained in a dry anomaly. The recurrent drying of the upper layers in spring and summer 2019 inhibited considerable infiltration, so that the moisture deficit of the lower soil layer persisted until winter 2019/20, when the relatively wet climatic conditions allowed a recharge of the lower soil layer moisture reservoir (Brakkee et al., 2022) and thus probably also of the groundwater (e.g., Brauns et al., 2020). Hence, the lack of soil moisture reached the entire soil column and thus the entire root zone of the vegetation during the summers of 2018 and 2019, placing the vegetation under soil moisture stress (Tijdeman and Menzel, 2021).



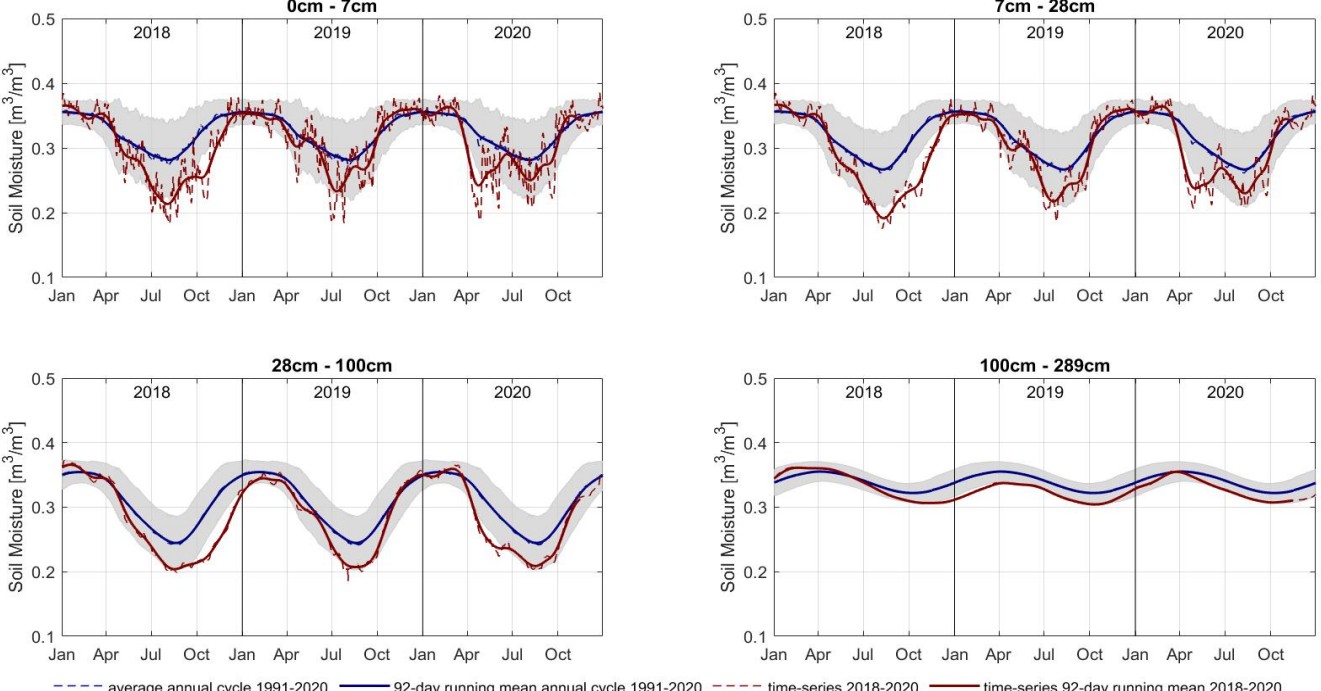

**Figure 4: ERA5 soil moisture in four different layers from surface (0 - 7 cm) to a depth of 2.89 m (100 - 289 cm) with two intermediate layers 7 - 28 cm and 28 - 100 cm depth. The red dashed line denotes the daily mean and the solid red line denotes the 92-day running mean in 2018-2020. The annual cycle of soil moisture with a daily resolution (dashed blue line) and the average running mean (solid blueline) are also shown. The grey shading indicates a range of ±1 standard deviation of soil moisture over the period 1991-2020 indicating the normal year-to-year variability of the soil moisture.**


### 3.1.5 Agricultural and hydrological drought of the year 2018

The abnormal soil moisture conditions are reflected in an abnormally low ET/PET ratio over the summer months (June, July, August), indicating a severe agricultural drought (Fig. 5, left). In almost the entire northern part of Germany, the agricultural drought index exceeds -2.5, which is equivalent to a return period of more than 160 years. However, the agricultural drought

is not limited to northern Germany but comprises large parts of central, northern, and northeastern Europe. The low soil moisture conditions also lead to a hydrological drought (low river flow) over the summer months (Fig. 5, right). However, the severity and spatial pattern of hydrological drought differs from the pattern of agricultural drought because propagation from soil moisture drought to hydrological drought takes time and typically leads to a lagged occurrence (Van Loon and Van Lanen, 2012) and a longer persistence (see section 3.1.3 on Surface water storage). Another reason is that hydrological drought can

spread along the river network affecting regions unaffected by low soil moisture (e.g., along the Danube river in eastern Europe). Nevertheless, in many parts of Germany and northern Europe, agricultural and hydrological drought coincided in the summer of 2018, affecting the possibility to irrigate as a means to alleviate the agricultural drought. This provides an example





of how co-occurring impacts (droughts) can amplify each other to cause even greater secondary impacts (agricultural yields, see section 3.1.6 on Impact on the agricultural production) in a similar way as co-occurring meteorological conditions trigger

CEs.

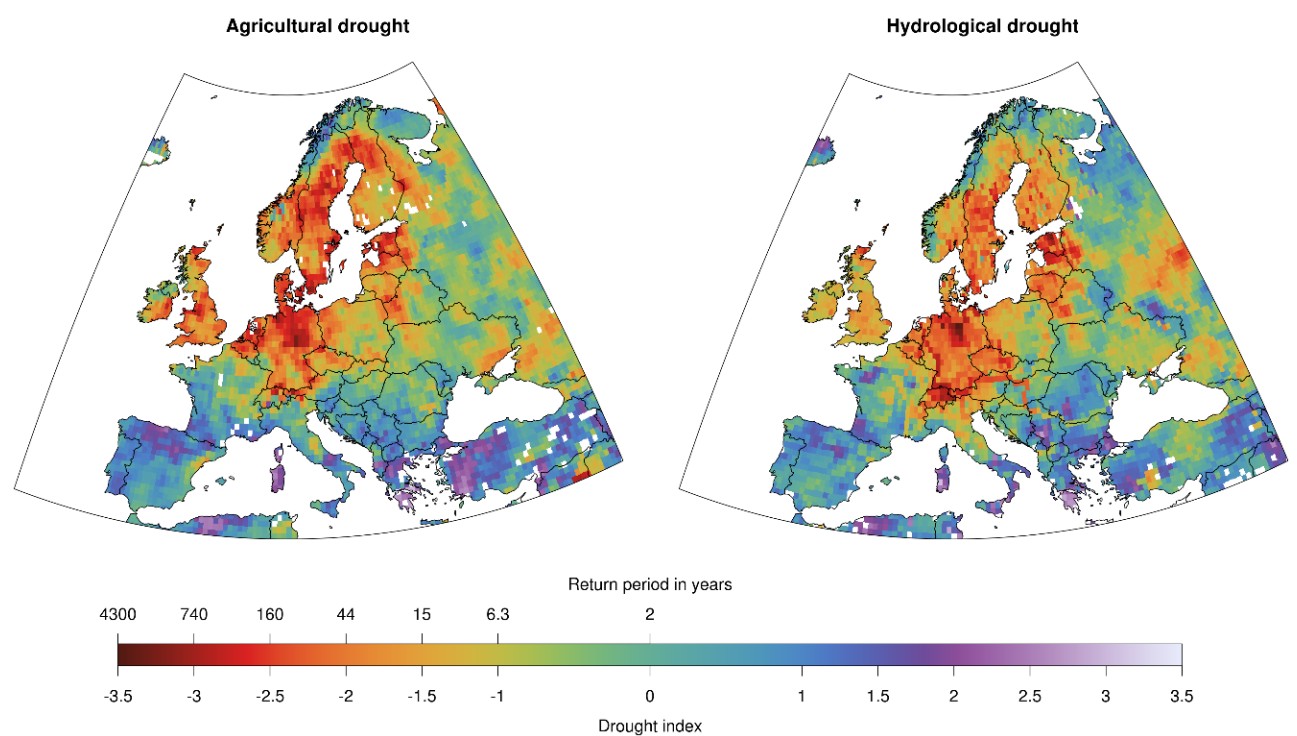

**Figure 5: Drought severity and return period of agricultural and hydrological drought during summer 2018. Note that drought severity (as expressed by the drought index) and return period are closely related (see Methods section 2.5).**

**3.1.6 Impact on the agricultural production of 2018**

In Germany, the hot and dry spring and summer of 2018 had an unprecedented impact on crop yields. Extremely low crop yields (Toreti et al., 2019a; Bellouin et al., 2020) led to large insurance claims over agricultural losses and financial support requests by farmers from governments in Germany (EUR 340 million), Sweden (EUR 116 million) and Poland (EUR 116 million) (D'Agostino, 2018; Munich Re 2019). Yields for winter wheat were more than 10% below the 30-year average and

13% below the previous three years. In some counties, yields were more than 40% lower than in previous years. Regionally, winter wheat was particularly hard hit in eastern and northeastern Germany, with an average loss of 22% compared to the last three decades. The HMD index and the SPEI indicate a severe heatwave and drought respectively, which was most pronounced in central and northeastern Germany. Figure 6 shows the explained variance of yield anomalies and the stress indices HMD, SPEI and CSI (see Methods section 2.6) revealing a strong connection between these components. However, not all regions



experienced such severe yield losses; winter wheat yield in southwest Germany was hardly affected, with losses of only 1.2% compared to the last three-decade average. A hydrological see-saw with rather wet conditions in southern Europe and the resulting yield increases characterise the unique combination of climatic anomalies in Europe in 2018 (Toreti et al., 2019a). Winter wheat productivity was even positively affected in some regions.

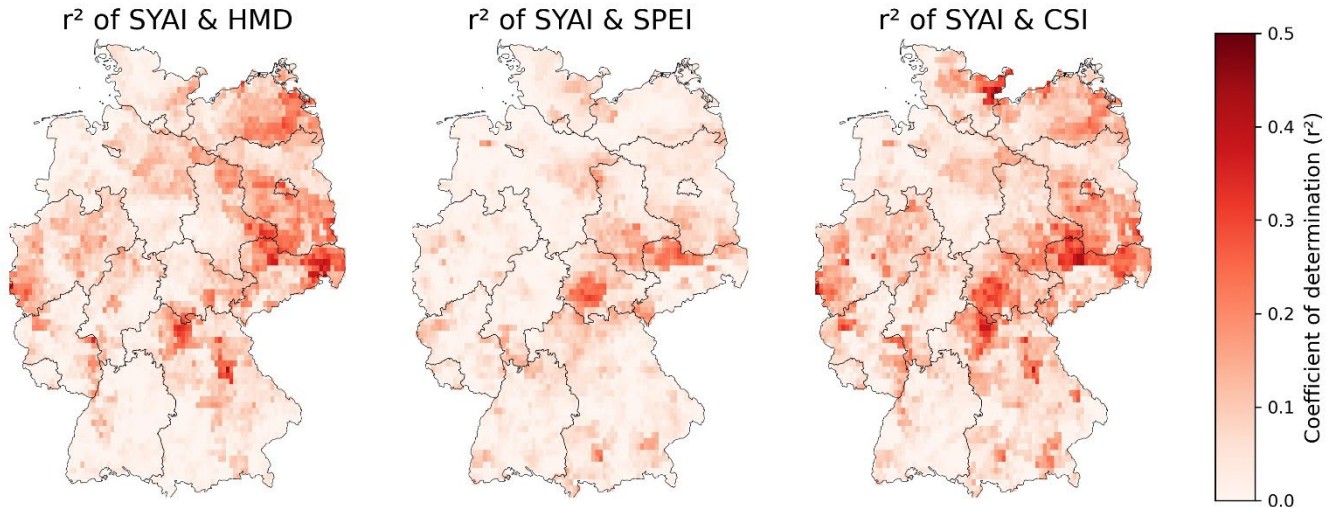

**Figure 6: Coefficient of determination of Standardized Yield Anomaly Index (SYAI) and stress indices Heat Magnitude Day (HMD), Standardized Precipitation Evapotranspiration Index (SPEI) and Combined Stress Index (CSI) demonstrate the impact of heat, drought and compound stress over the last three decades on winter wheat in Germany.**

### 3.1.7 Impact on the forests in 2018

The drought of 2018 was likely the largest source for severe forest disturbance in Europe for more than 170 years (Senf and Seidl, 2021), especially in central and northern Europe (Buras et al., 2020). Consequently, in the summer of 2018, about 11% of the central European forest area experienced early wilting (Brun et al., 2020), resulting in a large reduction in greenness (Schuldt et al., 2020; the three aforementioned studies are based on NDVI data). The 2018 drought continued into 2019, making the consecutive European droughts of 2018 and 2019 unprecedented in the last 250 years (Hari et al., 2020). The low soil

moisture content in 2018 and an increased water-vapour pressure deficit in the following two years were the main drivers for the forest disturbances of about $4.74 \times 106$ ha in Central Europe (Senf and Seidl, 2021). The likely cause for these forest damages was that trees under drought and heat stress experience carbon starvation (Bastos et al., 2020) and have risk for embolism, which causes failures in water transport (Allen et al., 2015; Schuldt et al., 2016). The drought and heatwave in that period facilitated outbreaks of bark beetle, enhancing the damage levels to forests. As such, insect outbreaks in Central Europe

had a 2-3-fold increase in annual losses between 2017 and 2018 (Hlásny et al., 2019) and extraordinary mortality and damage occurred during 2018 in Sweden due to rapid beetle population growth (Öhrn et al., 2021). Although wildfires have decreased



on a global scale recently, Central Europe is likely to face larger and more frequent forest fires (Feurdean et al. 2020, Milanovic et al. 2020; Carnicer et al., 2022), which can have severe environmental, economic and social consequences (Lidskog et al. 2019).

## 3.2 Precipitation-wind compound events during 2018


In this section, CEs that involve heavy precipitation and strong winds are described. Examples include two January windstorms (Friederike and Burglind) and several weeks of convective activity in May and June of 2018.

### 3.2.1 Loss and damage of compound vs. non-compound wind extreme events of the winter 2018

On January 16, windstorm Friederike formed as a low-pressure system near Newfoundland. Within the next two days,
Friederike intensified and quickly travelled across the Atlantic (Fig. 7), losing its closed structure at 1800 UTC on 17 January. Friederike re-intensified over the British Isles on 18 January while crossing the jet streak towards the northern jet exit region, a behaviour favourable for intense windstorm development (e.g., Pinto et al., 2009). The storm moved eastward over the North Sea, Germany and Poland, and weakened after 19 January over eastern Europe (Fig. 7a). Analysing the compound character of Friederike around peak intensity using the hourly ERA5 reanalysis data (Fig. 7b,c), the typical near-surface wind and
precipitation structure of intense extratropical cyclone is found (e.g., Dacre et al., 2012). Strong 10 m wind gusts (maximum values of 34 m/s relative to the Earth's surface) were present behind and to the right of the eastward moving cyclone centre. Heavy precipitation occurred at both the warm front to the northeast of the centre, wrapping around as the cyclone approached its mature stage, and along the east-southwest stretching cold front (Fig. S1a). During the 12 h period when Friederike passed through Germany from 0600 UTC to 1800 UTC on 18 January, the persistently active warm front left a widespread footprint
near the northern edge of the cyclone centre (Fig. 7b) with ERA5 accumulated precipitation exceeding 17 mm. Meanwhile, the cold front contributed to a high precipitation rate (> 4 mm/h based on ERA5) along a narrow west-east oriented band across northern France and southern Germany (Fig. 7c). The co-occurrence of strong winds and heavy snowfall gave to this storm the risk and damage characteristics of a CE (Fig. 7b,c). The highest damages were reported in Ireland, Great Britain, northern France, Belgium, the Netherlands, Germany, Czech Republic and Poland, where gust measurements suggested wind speeds of
the order of 100 – 150 km/h. At higher altitudes the observed wind gusts reached 173 km/h at the Sněžka in Czech Republic and 203 km/h at Brocken in Germany (Haeseler et al., 2018). Wind and snowfall associated with Friederike caused further traffic disruption, power outages, property damage including falling trees, and several deaths. Friederike was the strongest storm affecting central Germany since windstorm Kyrill in 2007.

Another CE affecting Germany in the same month was the windstorm Burglind, which formed on the 2nd January 2018. The
depression intensified rapidly as it moved eastwards towards the British Isles (Fig. 7a). It reached a peak intensity of 968.9 hPa at 0600 UTC on 3 January 2018 over the North Sea, followed by a weakening over the Baltic Sea. The long active cold front affected a large area of western Europe (Fig. S1b). Heavy precipitation with daily values > 30mm led to rapid snowmelt and massive flooding in many regions. Around the time of the peak cyclone intensity, widespread areas were simultaneously



affected by high precipitation intensities (> 4 mm/h) and high wind gust (~100km/h) (Fig. S1c,d). Further detailed information

on Burglind can be found in Eisenstein et al. (2022, see their section 5). Compared to storm Friederike, the compound features

of Burgling were more strongly shaped by orography.

**Figure 7: (a) Cyclone tracks of windstorms Friederike (black) and Burglind (blue) in January 2018. Big circles show locations at 0000 UTC on the day as indicated, with the central pressure noted below. Red circles indicate their lifetime peak intensity based on**

**the minimum pressure. (b) Mean sea level pressure (thick contours; increasing from 960 hPa with 5 hPa intervals) at 1200 UTC on 18 January 2018 (location of Friederike shown by the star) and maximum precipitation intensity (shaded) during the period 6 h before and after (locations shown by black circles). (c) Same as b, but for the wind gust at 10 m height (shaded). All fields are derived from the ERA5 reanalysis (Hersbach et al., 2018).**





Although the co-occurrence of extreme wind and precipitation is discussed in previous studies for specific events (e.g. Fink et al., 2009 for storm Kyrill) or globally (Martius et al., 2016; Messmer and Simmonds, 2021), there are no studies so far quantitatively evaluating the effect in terms of loss damage. To distinguish between single extreme wind speed events and compound extreme wind speed and precipitation events, we follow the definition of Martius et al. (2016), where both variables are considered simultaneously (see the Methods section 2.7). The loss damage distribution for compound and non-compound

events determined from loss data of the GDV is depicted in Fig. 8. For Friederike, there are three days where a co-occurrence of wind and precipitation extremes can be identified over Germany, i.e. 17, 18, and 19 January 2018. The loss ratio for these three days is marked with blue dots in the right column of Fig. 8. The GDV Naturgefahrenreport (2019) reports 900 Million € loss damage for Germany with respect to Friederike. To this date, it was the most damaging winter storm of the last ten years.

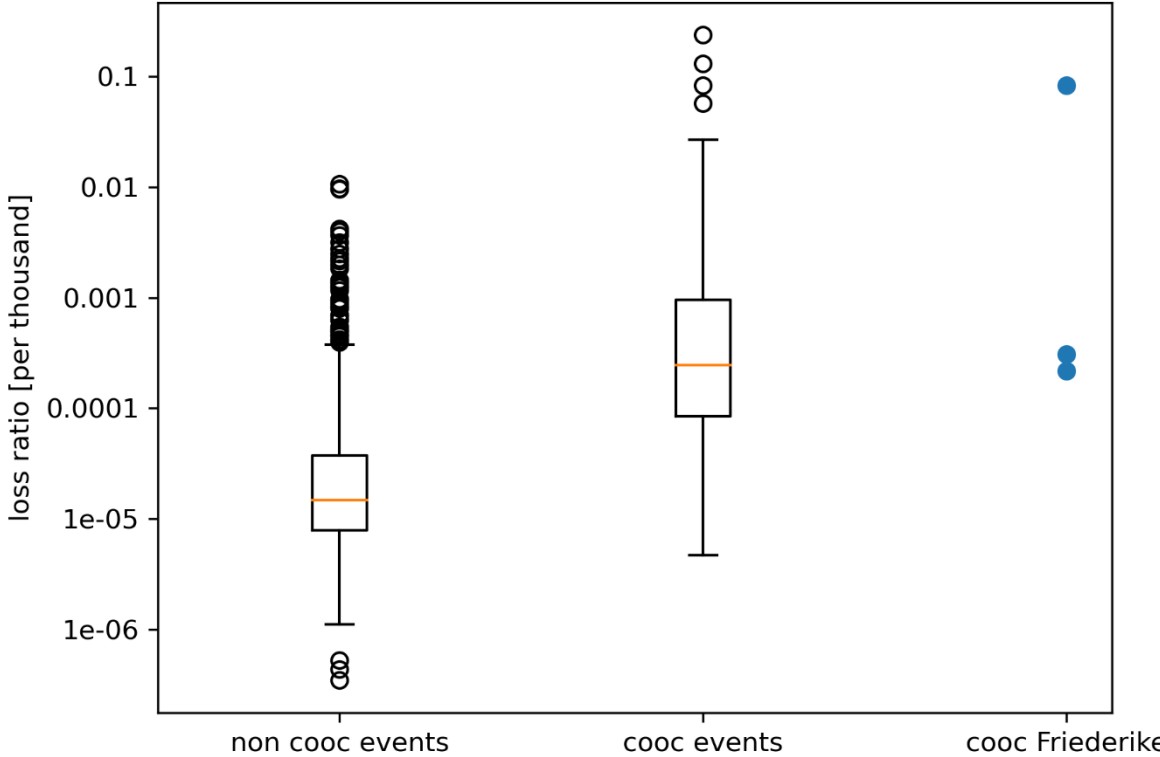

**Figure 8: Loss ratio of residential buildings [per thousand] accumulated over Germany for winter events from 1997-2016. Each dot represents one day. Left (non cooc events) bar shows all days which cannot be linked to co-occurrence of both extreme wind and precipitation, i.e. single extreme events. Middle bar (cooc events) shows all days which can be linked to co-occurrence of extreme wind and precipitation. Right bar shows the three days for the co-occurrence during windstorm Friederike. The loss ratio is defined by loss normalized with the local sum of insured values.**




### 3.2.2 Concurrent heavy rain and storm extremes – estimation of probability of event occurrence

In a detailed analysis of the probability of co-occurrence of extremes, based on copulas (see Methods section 2.8), the annual maximum values of hourly precipitation and wind speed data at the station Münster/Osnabrück Airport of DWD that are available since 1996 were analysed. The records show an increase in the intensity and frequency of the variables but lack a
statistically significant trend. The occurrence probabilities for concurrent precipitation and wind extreme events are shown by the black isolines in Fig. 9. The grey dots are pseudo-observations (artificial precipitation and wind combinations generated using the copula function), while the black, red and green dots mark the observed CEs at the station for each year and the green dot represents Friederike. The distribution of the dots illustrates that - depending on the precipitation duration – wind or precipitation, individually, may not be extreme. A counterexample is the year 2020 windstorm Sabine (red dot; known as
Ciara) for which the simultaneous wind and hourly precipitation values both correspond to the respective annual maximum event of that year. The location of the dots relative to the isolines define the return period of the event, where the return period of windstorm Friederike at the station Münster/Osnabrück exceeded 5 years and the one of windstorm Sabine 100 years.

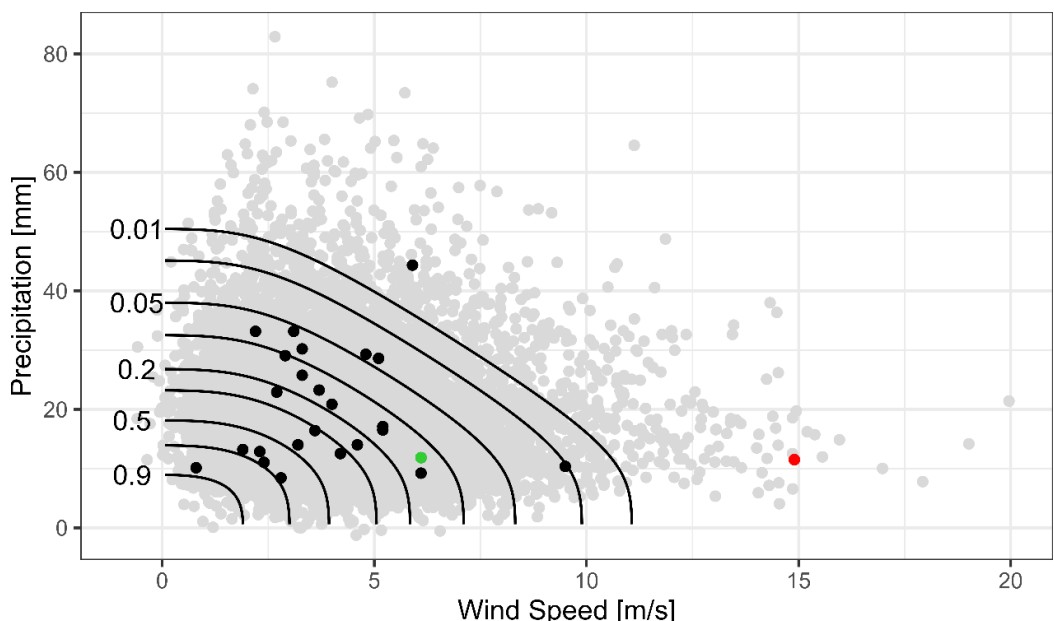

Figure 9: Multivariate analyses for the temporal compound event heavy precipitation and strong wind determined with Copula
functions: quantile isolines (lines of equal probabilities), observed event combinations (black dots) and the pseudo-observations (grey dots). Green dot represents storm Friederike (2018) and red dot storm Sabine (2020).



### 3.2.3 Rockfall events

Another hazard with CE triggers observed in connection with Friederike was a rockfall event. Although wind is normally not
considered as one of the triggering factors (D'Amato et al., 2016), such events may also occur as the consequence of the
precipitation associated with the windstorm. It is well known that heavy precipitation can initiate landslides and rockfall events.
Slope susceptibility is influenced by pore water/fissure water preconditions, rendering them events with multivariate and
temporally compounding triggers. With respect to rockfall, another potentially triggering factor are freeze-thaw cycles prior
to the event. In terms of reported hill slope failures, storm Burglind was more effective than storm Friederike. For storm
Friederike, only one rockfall event is registered in the landslide database for Germany. The event occurred near Göttingen in
Lower Saxony. Figure 10 shows the relationship between the predictors (across-site percentile of a fissure water proxy D -
precipitation minus potential evaporation determined for the last 5 days-, the local percentile of daily precipitation and the
binary information if a freeze-thaw cycle occurred within the last 9 days) and the rockfall probability expressed as percentage
change with respect to the climatological probability. The red dot indicates the conditions on the 18 January 2018 in the area
of the event associated with Friederike. The soil at the location was still wet after storm Burglind at the beginning of month
and the fissure water (proxy D) at its 83rd percentile. The daily precipitation on the day of the event was 4.5 mm (REGNIE
data, Rauthe et al., 2013), which corresponds to the 77th percentile for the given location, assuming immediate melting of the
reported snow due to the above freezing air temperature at the event location. The probability of a rockfall event was further
increased by pre-event thawing conditions. The logistic regression model suggests that the probability of rockfall on that day
was increased by almost 250% (3.5 times more likely) compared to the long-term climatology.





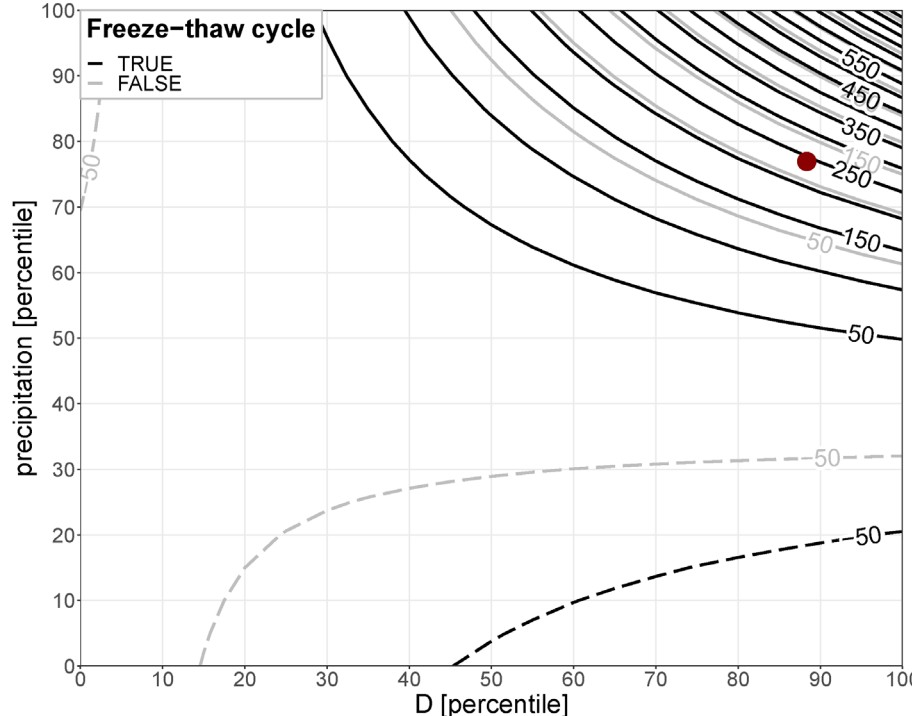

**Figure 10: Rockfall probability expressed as percentage change with respect to the climatological probability (isolines) as a function of moisture preconditions (D), daily precipitation and preceding freeze-thaw cycles. The red dot marks the conditions in the vicinity of Göttingen (Lower Saxony) at 18th January 2018.**

### 3.2.4 Convective cluster events of the summer 2018

The spatially as well as temporally compounding nature of severe convective storms (SCSs) can be demonstrated on the example of a three-week series of SCSs in western and central Europe from 22 May to 12 June 2018 leading to the unusual high temporal accumulation of CCEs lasting several days (Fig. 11). During this period, an exceptional persistence of reduced stability combined with sufficient moist air masses caused high thunderstorm activity daily in France, Belgium, Netherlands, Luxembourg, Germany, Switzerland, and/or Austria, associated with precipitation accumulations of up to 80 mm/h within 1 hour and several flash floods (Mohr et al., 2020). The temporal compounding nature of the serial clustering of SCSs over several days to weeks over the same geographic region may increase the probability of flooding and damage. Figure 11b shows a large amount of identified CCEs, especially those with large spatial extent (> 5000 km2), during a three-week period over western and central Europe indicating high thunderstorm activity whose accumulation was unusual (Piper et al., 2016; Mohr et al., 2020). The repeating occurrence is caused by persistent synoptic conditions that favour thunderstorm development over several days to weeks. The presence of atmospheric blocking has been found to be highly conducive to such prolonged thunderstorm episodes, which typically occur on its western and /or eastern flanks (Piper et al., 2016; Mohr et al., 2020; Kautz et al., 2022). Based on statistical analyses, Mohr et al. (2019) found that a block over Scandinavia or over the Baltic Sea



favoured the occurrence of thunderstorms in western and central Europe along the western flank of the blocking system due
to of southwesterly advection of warm, moist, and unstable air masses. It is expected that low-frequency modes of climate
variability like North Atlantic Oscillation (NAO) or East Atlantic Pattern (EA) could also have an impact on clustered
thunderstorm activity over several days (Piper et al., 2019), as these patterns are connected with atmospheric blocking.



**Figure 11: (a) Lightning strokes on the 28 May 2018 in western and central Germany. Each colour represents a convective cluster**
**event (CCE) resulting from the ST-DBSCAN method (see Methods section); black dots represent lightning strokes identified as**
**noise. (b) Daily number of CCEs between the thunderstorm episode from 22 May to 12 June 2018. The colours represent CCEs of**
**different sizes. Note: clusters that overlap are separated in time.**



## 4 Compound events under climate change

In this paper, we have analysed in detail several extreme events that have affected Europe within the calendar year of 2018,
starting with the windstorm series in January, followed by a period of heavy thunderstorms in May/June and extended heatwave
in July and August which affected various parts of Europe, and the associated drought effects extended well into the autumn
season. Our analysis clearly revealed the multi-variate and complex characteristics of the events, and thus they can undoubtedly
be classified as CEs. Within climXtreme, the analysis of compound variables was used as a tool to investigate the impacts of
man-made climate change. For the two overarching compound types collected in this contribution (hot and dry; wet and windy)
we now analyse possible changes of their frequency of occurrence under future climate conditions. Recent studies have
provided evidence that regionally extreme hot and dry conditions such as the summer 2018 are expected to become more
frequent in the future (e.g., Toreti et al. 2019a; Zscheischler and Fischer, 2020; Aalbers et al., 2023; van der Wiel et al., 2021;
Bevacqua et al., 2023). Figure 12 shows a comparison of the occurrence of CEs under recent (1975-2025) climate conditions,
with a global mean surface temperature of 1 °C above pre-industrial levels, and under future (2025-2075) conditions of 3 °C
above pre-industrial levels, based on the 30-member CMIP6 MPI-GE under SSP585 (see Methods section 2.11). For drought
and heat events, our analysis reveals a clear increase in both the frequency and intensity of extreme compound heat and drought
years (Fig. 12, left). Over the past 50 years, extreme compound heat-drought events have occurred with a probability of 1.5%,
or about 1-2 times per century. Over the next 50 years, such extreme CEs are projected to become almost 10 times more
frequent, occurring more than once every 10 years, and reaching much higher temperatures and precipitation deficits.

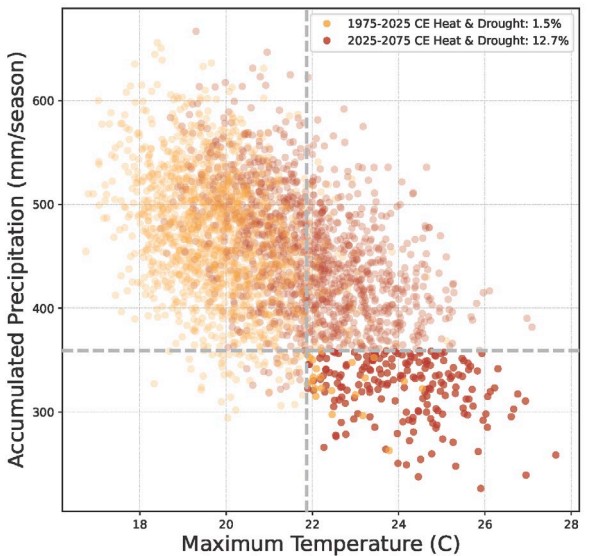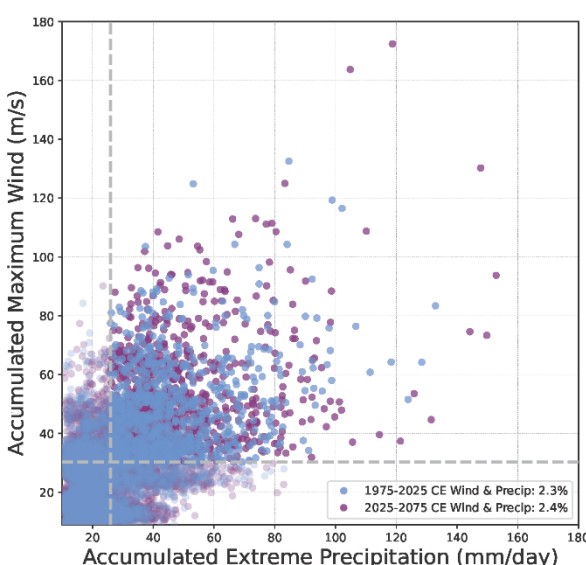


**Figure 12: Changes of temperature-precipitation (heat-drought, left) and precipitation-wind (wet-windy, right) compound events
occurrence in terms of frequency and intensity, at 1 °C (1975-2025) and 3 °C (2025-2075) above the pre-industrial levels based on
the 30-member CMIP6 MPI-GE under SSP585 (Olonscheck et al., 2023) over Germany.**



The likelihood of winters with extreme compound precipitation-strong wind events does not change significantly with global
      warming in the CMIP6 MPI-GE projections (Fig. 12, right). Such wet and windy winters, where both precipitation and wind
      are extreme, are projected to occur about once every 50 years. However, although the number of projected events remains
      roughly the same, the intensity of the actual wind and precipitation levels reached during the most extreme compound wet-
      windy events increases substantially in the near future. Moreover, the unprecedented character of the drought conditions of the
events in 2018 poses challenges to the ability of climate models to realistically reproduce such conditions.

      A factor that strongly modulates the occurrence of extremes in Europe is the soil moisture availability, as decreasing soil
      moisture availability initiates a suit of processes feeding back into an intensification of both heat waves and droughts (Miralles
      et al., 2019). Figure 13 presents a comparison between the observed (ERA5), simulated (historical) and projected (RCP8.5)
      drought conditions via the SPEI moisture availability index (see Methods section 2.12). For the historical period 1979-2019,
the Reanalysis shows trends in the 3-year running mean SPEI, with a clear drying tendency in central and southern Germany
      during the warm season, with lower values elsewhere in Germany (Fig. 13a). Part of this trend may be related to the multi-
      year drought of 2018-2020 at the end of the time series. The CMIP5 multi-model ensemble under observed (historical)
      greenhouse gas concentrations (Fig. 13b) fails however at depicting the trends for the past decades, while Euro-CORDEX
      (Fig. 13c) simulates an increased water availability across Germany. The future projections following the RCP8.5 scenario are
roughly consistent between CMIP5 (Fig. 13d) and EURO-CORDEX (Fig. 13e). Both ensembles predict a considerable trend
      towards more impactful multi-year drought conditions under RCP8.5. Differences between the various sources of data may be
      related to specific characteristics and settings of the climate models, such as the treatment of anthropogenic aerosols, the
      inherited uncertainty and bias of climate models to replicate the precipitation variability at the regional and local scale, the
      differences of the convective precipitation during the warm part of the year (Dyrrdal et al., 2017), but also the use of multi-
model ensembles that may mask the individual model skill (Ridder et al., 2022). These examples demonstrate the need for
      further model development to improve their ability to accurately reproduce observed CEs and their characteristics, thus
      reducing the uncertainty of future projections and contributing to improved prevention, risk management and future
      preparedness.





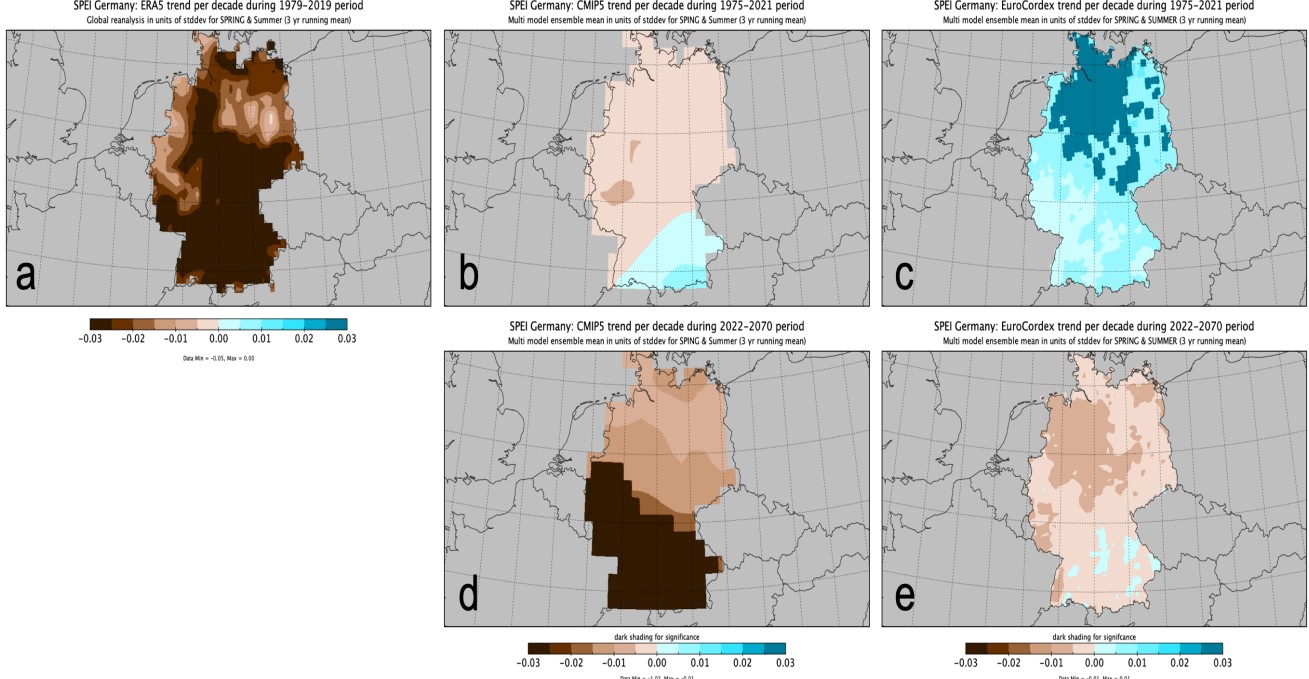


**Figure 13: Decadal trends, 1975-2021 and 2022-2077, of the Standardized Precipitation Evapotranspiration Index for (a) ERA5, (b) historical CMIP5 multi-model ensemble, (c) EURO-CORDEX multi-model ensemble, (d) RCP8.5 CMIP5 multi-model ensemble, and (e) RCP8.5 EURO-CORDEX multi-model ensemble. Significant model ensemble grid points are shaded dark.**

## 5 Conclusions – Lessons learned and future steps

The case studies presented in this paper illustrate the role of CEs for impact studies exemplified with the calendar year of 2018, as an exceptional year of various multi-variate extremes. Our approach follows a user-oriented climate change research path which addresses the socio-economic effects of the meteorological changes (IPCC, 2022). Within climXtreme, a variety of statistical approaches and datasets have been explored and implemented. The showcases presented in this paper include multivariate, pre-conditioning, temporally and spatial CEs (see Zscheischler et al., 2020; Bevacqua et al. 2021). Combinations of heat and drought stress as well as windstorms, convective storms and heavy precipitation led to unprecedented impacts on ecosystems, infrastructure and societies. These case studies demonstrate science advancements in various aspects and contribute to a better process understanding of CEs, as well as generated new knowledge with respect to better informed risk management, enhanced preparedness and resilience against CEs, their consequences and impacts at the German national and European scales.

The hot and dry summer of 2018 was characterised by a combination of concurrent meteorological conditions and impacts that amplified each other and led to even larger secondary impacts with unprecedented social, ecological and economic



consequences. The heatwaves of summer 2018, prominent events in terms of seasonal variability, were preceded by an intense and persistent drought that began in spring and extended into early summer and autumn. The persistent large-scale atmospheric

blocking conditions favoured the occurrence of the heatwaves over central Europe, with hotspot areas coinciding with regions of weak winds between the polar and subtropical jets. The soil moisture deficit that developed in spring and early summer 2018 reached the lowest soil layers with a time lag of about three months, while the intense dryness affected groundwater storage. Taking into account the memory effect of groundwater storage, a full recovery from a drought affecting surface water storage takes longer than one year. A partial recovery of deficits during the winter of 2019 did not reach the lowest soil layers,

which were affected anew by the recurrent dry conditions of the following spring and summer. The lack of soil moisture extended to the entire soil column and thus to the entire root zone of the vegetation during both summers of 2018 and 2019, placing the vegetation under soil moisture stress. In addition, the ability to use irrigation as a tool to mitigate the agricultural drought was further compromised by the co-occurrence of the agricultural and hydrological droughts. The constellation of climatic anomalies across Europe resulted in an average winter wheat yield loss of 22% compared to the previous 30 years

over large areas of Germany. Lastly, the combination of drought and heat stress on trees led to a sharp reduction in green cover, insect outbreaks and an increased risk of forest fires in Europe, with serious environmental, economic and social consequences.

The combined influence of strong winds and heavy precipitation, associated with a series of intense storms in the winter of 2018, demonstrated the risk and damage characteristics of CEs. Among these, Friederike (David) and Burglind (Eleanor) were

connected with strong winds and heavy snowfall that caused extensive damage along their paths across Europe. The assessment of the impact of winter storm Friederike in terms of losses clearly classifies it as a CE, making it the costliest windstorm of the last decade. The analysis of the wind and precipitation variables during the winter storm Friederike showed the extreme nature of the wind, and the long duration of the precipitation, related to the slow movement of the storm. Furthermore, the sequence of the two storms may have amplified the effects of the long precipitation and thaw levels, favouring the rockfall event in

Lower Saxony. In the early summer of 2018, the developed atmospheric blocking not only contributed to the occurrence of the heatwaves, but also favoured the occurrence of severe convective storms on the western flank of the block over several days (up to three weeks) with unusually high precipitation. The impact of such persistent convective events increases the risk of flooding and damage, especially following the intense drought in spring 2018 in the area.

Our analyses have also shown that both types of CEs may change under future climate conditions both in terms of their

frequency and severity. Indeed, future projections indicate an increase in the frequency of many types of extreme events (IPCC, 2022), so quantifying the likelihood of future extreme events is becoming increasingly important for adaptation planning. Furthermore, it is critical to accurately examine the links between these types of events, as the risk and return periods of extreme events may be significantly underestimated by assuming the independence of these events, or by examining only a single extreme event. To ensure a coherent risk assessment of high-risk events such as CEs, multiple drivers should be

considered that play a synergistic and reinforcing role. Further studies aim at expanding the current knowledge on the complex relationships between CEs and large-scale fields at different time horizons in order to improve the detection and thus the



understanding of the climate system. The next step would be to validate the physical relationships between predictors and CEs to identify the physical mechanisms that drive or impact these events, which may improve the analysis and potentially the prediction of CEs. Furthermore, given the nature of CEs, their complexity and their direct links to impacts, a definition of events that relies primarily on the relevant impacts prior to their statistical characterisation is required. Considering the uncertainties associated with future projections and the multivariate character of drivers and events, CEs pose major challenges and lead to limitations in the development of effective adaptation strategies, making research in this direction imperative. Therefore, further dedicated research is needed towards a comprehensive view of CEs and their impacts in a warming world.

**Code availability**

Code is available from the authors upon request.

**Data availability**

The ERA5 (https://doi.org/10.1002/qj.3803, Hersbach et al., 2020) reanalysis data are publicly available via the Copernicus Climate Change Service (https://doi.org/10.24381/cds.adbb2d47, Hersbach et al., 2023). The gridded observational datasets E-OBS (https://doi.org/10.1029/2017JD028200, Cornes et al., 2018) are publicly available on the European Climate Assessment & Dataset website (https://www.ecad.eu/download/ensembles/download.php, ECA&D, 2023). The observational datasets (https://doi.org/10.5194/asr-10-99-2013, Kaspar et al., 2013) and the phenological data from the German Weather Service (DWD; https://doi.org/10.5194/asr-11-93-2014, Kaspar et al., 2015) are publicly available on the DWD website under their Open Data Portal (https://opendata.dwd.de/, DWD, 2023). The yield productivity data (http://dx.doi.org/10.22029/jlupub-7177, Ellsäßer and Xoplaki, 2022a), the yield anomaly catalogue (http://dx.doi.org/10.22029/jlupub-7176, Ellsäßer and Xoplaki, 2022b) and supplementary data (http://dx.doi.org/10.22029/jlupub-7203, Ellsäßer and Xoplaki, 2022c) are publicly available at the JLUpub research data repository. Historic climate data from the GSWP-W5E5 dataset used for LPJmL5 simulations are available from https://doi.org/10.48364/ISIMIP.982724 (Lange et al., 2022). The historical data of atmospheric N deposition and atmospheric $CO_2$ concentrations can be obtained from https://doi.org/10.48364/ISIMIP.600567 (Yang and Tian, 2020) and https://doi.org/10.48364/ISIMIP.664235.2 (Büchner and Reyer, 2022), respectively. All input data, model code, model outputs, and post-processing scripts that have been used to produce the LPJmL-related results in this paper are archived at the Potsdam Institute for Climate Impact Research and are available upon request.

**Competing interests**

At least one of the (co-)authors is a member of the editorial board of Natural Hazards and Earth System Sciences. The peer-review process was guided by an independent editor, and the authors have also no other competing interests to declare.



**Author Contribution**

EX and FE coordinated the interdisciplinary task force on compound events within climXtreme and this collaborative paper, conducted the agriculture case study analysis with Figure 6, prepared the first and final drafts of the paper based on the contribution of the co-authors; ER contributed the drivers of the hot summer of 2018 case study with Figure 1; SS contributed the detection of spatial patterns of extreme events and Figure 2; LG and SK contributed with the surface water storage case study analysis and Figure 3; LJ contributed the soil moisture case study and Figure 4; JH contributed with the analysis of the

agricultural and hydrological droughts and Figure 5; DG and FK contributed with the forestry case study; JGP and T-CC contributed the windstorms description and prepared Figure 7 and S1; JGP contributed to the compound events under climate change section; JG conducted the loss analysis for windstorms with Figure 8 and contributed to the methods section; FS conducted the copula analysis and prepared Figure 9; KNM contributed the rockfall events case study analysis with Figure 10, drafted the section on precipitation-wind compound events and contributed to the compound events under climate change

section; SM and MA analysed convective storms and provided Figure 11; LSG did the CMIP6 MPI-GE projection study and prepared Figure 12; KH contributed the analysis of soil moisture representation and trend in model simulations with Figure 13. NL and OV contributed to the temperature-precipitation compound events section; JL contributed drafting different sections and versions of the paper and all authors followed the analysis from the beginning, provided text and edited/commented the final version of the manuscript.

**Acknowledgements**

This paper is a collaborative effort within the BMBF climXtreme project, for which the authors acknowledge funding (grant numbers 01LP1901A, 01LP1903C, 01LP1901F, 01LP1903A, 01LP1902J, 01LP1903F, 01LP1902M, 01LP1901E). EX, NL, OV acknowledge support by the EU Horizon 2020 Project CLINT under Grant Agreement number 101003876. EX acknowledges support by the BMWK project DAKI-FWS (grant number 01MK21009I) and the EU Horizon Europe project

MedEWSa under Grant Agreement number 101121192. JGP thanks the AXA research fund for support. LSG has also received funding from the European Union's Horizon Europe Framework Programme under the Marie Skłodowska-Curie grant agreement No 101064940. SMVB acknowledges funding from the DFG training group NatRiskChange (grant No GRK 2043/1). We acknowledge the World Climate Research Programme's Working Group on Coupled Modelling, which is responsible for CMIP, and we thank the climate modelling groups for producing and making available their model output. For

CMIP the U.S. Department of Energy's Program for Climate Model Diagnosis and Intercomparison provides coordinating support and led development of software infrastructure in partnership with the Global Organization for Earth System Science Portals. We acknowledge the World Climate Research Programme's Working Group on Regional Climate, and the Working Group on Coupled Modelling, former coordinating body of CORDEX and responsible panel for CMIP5. We also thank the climate modelling groups for producing and making available their model output. We also acknowledge the Earth System Grid

Federation infrastructure an international effort led by the U.S. Department of Energy's Program for Climate Model Diagnosis



and Intercomparison, the European Network for Earth System Modelling and other partners in the Global Organisation for Earth System Science Portals (GO-ESSP). We acknowledge the E-OBS dataset from the EU-FP6 project UERRA (http://www.uerra.eu) and the Copernicus Climate Change Service, and the data providers in the ECA&D project (https://www.ecad.eu).

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
