# Peer review of "Compound events in Germany in 2018: drivers and case studies"

_EGUsphere, 2023_

## Author Response (AR1)

**Compound events in Germany in 2018: drivers and case studies**

**General response**

We extend our gratitude to the two anonymous reviewers for their careful reading and constructive criticism of our article. the reviewers' comments will significantly improve the quality of the manuscript. We have carefully considered the reviewers' comments and we respond accordingly to each of those. Updates to the
Detailed responses are provided below. The reviewers' comments are in black color, while our responses are in blue. All line numbers in the response documents refer to the lines in the revised manuscript without Track Changes.

**Reviewer 1**

*Comment:*
The manuscript describes a set of specific analyses done to understand and characterize complex extremes, such as the ones occurred in 2018. The text is well written and results clearly discussed. Despite the great readability, methods should be better explained and more details provided. This would give all readers the possibility to better appreciate the main findings and the figures. This is especially true for sections 2.2, 2.3, 2.8. For instance: the EPI and the TPDM are barely described; the choice of the ensemble size of the hydrological simulations is not discussed, neither is the experimental setting.

Thank you very much for your constructive feedback on our manuscript.
We appreciate your suggestion to provide more detailed explanations of the methods, as this indeed enhances overall clarity and allow readers to better understand and appreciate our findings. In response to your comments, we expanded sections 2.2, 2.3, and 2.8 with more detailed descriptions, particularly regarding the EPI and TPDM methodologies. We also provide a clearer discussion of the rationale behind the ensemble size chosen for the hydrological simulations and give a more explanation of the experimental settings. The hydrological simulations and settings are taken from Hartick et al. (2021) which is mentioned in the paper, and details can be found therein.

**Reviewer 2**

Point-to-point response to Reviewer #2:

*Review of "Compound events in Germany in 2018: drivers and case studies" by Elena Xoplaki and colleagues.*

*This manuscript sets out to discuss compound events on the example of Germany in 2018. It is reasonably well-written and structured. A number of very long sentences make it hard to follow at times and the structure of the methods could be improved.*

*I acknowledge the effort that went into this work and I'm reasonably sure that it will be a good resource to look up European extremes in 2018 and case studies. But I also feel that the authors could have done a better job at building a more concise (i.e., shorter) story and make the paper feel less like a loosely connected collection of results in particular given the topic of compound events.*

*I do not expect the authors to overhaul the entire paper but I would encourage them to include some more connecting and structuring elements like tables or flow charts summarizing the methods and comparing the different results and their relation.*

*My only major issue is the almost complete lack of conclusions in the conclusion section. One thing that might be good to address would be: What did we learn from this study that we did not already know (apart from collecting it in one place).*

Thank you very much for your valuable and insightful comments on our manuscript. We are grateful for your recognition of the effort that went into this work and your positive outlook on its potential as a resource for understanding European extremes that happened in 2018.

We fully acknowledge your suggestions regarding the length and structure of the manuscript. We carefully reviewed the text to identify and shorten lengthy sections and complex sentences to improve readability. We also made the narrative more concise and focused, ensuring that the paper presents a cohesive story rather than a collection of loosely connected results. To enhance clarity, we incorporated more information that make the link between the different sections more clearly.

Due to the constraints on length, we would prefer not to include additional tables or flow charts. We believe the clarity and structure of the additional text are sufficient to convey the necessary information without these elements.

We revised this conclusion section emphasizing the new insights gained from the various events in 2018 and their importance within the context of compound events.

**Specific comments**

*49: switch to past tense?*

We have harmonised the use of past tense along the whole manuscript.

*71: "Zscheischler et al. (2018) further defined CEs as combinations of events that are individually not necessarily extreme, or multiple drivers and/or hazards, but in combination often lead to disproportionate impacts on people and ecosystems"*

*How is this different from line 68?*

*"3) a combination of events that are not themselves extremes but lead to an extreme event or impact when combined."*

68: We revised this section to eliminate the repetition and enhance the clarity of the content.

*74 "To quantify the probability of CEs in today's and future climate is of great importance specifically adaptation planning for" → "for adaptation planning for"?*

70: We made the change.

*105: This is just an opinion and I appreciate that the authors also want to advertise their project with this paper but maybe they could tune down the references to climxtreme a bit? At least I can not see why it is relevant to the reader how climXtreme organized the investigation underlying this manuscript and the manuscript is already very long.*

*Also 134, 613, …*

Thank you for pointing out the overemphasis on the climXtreme project. We put less emphasis on the project and deleted related text.

*Section 2: I am sorry but find this section quite hard to follow and at the same time it probably does not provide enough information to actually understand what was done in detail. I appreciate that it is not easy to write a concise method section for such an extensive paper but I think this could be done better. Currently the reader is thrown into more than 10 sub-sections mixing individual extreme definitions, compound events, and impacts. In addition, the level of detail somewhat differs between sections and there are many design choices that are not really reflected or compared.*

The challenge we face, as has been brought up by the reviewer, is that we have different case studies and storylines that address different extremes and compounds including different definitions, processes, modelling, including various time and space scales. We revised some of the parts of section 2 to make them more clearly. We refer also to our answer to reviewer 1 who also suggested clearer formulation of some of the sections.

*Just two examples:*

*- the authors repeatedly state that the summer months JJA are used (lines 157, 199, 272) but in other sections they only refer to summer (145, 161) or other periods such as the hydrological year (161) or the warm season (282). Given that this study is about compounding it seems important to be clear on the properties of the different events considered and their relation.*

*- There are a range of relative thresholds used to define extremes. I fond some based on the 90, 95, and 98 percentiles. I'm not saying that this is inherently problematic but it should be made clear.*

*My suggestion would be to start with an overview section explaining the structure of this section. This could include a table summarizing the main features of each analysis to ensure a similar level of detail and ease comparing differences.*

As mentioned above, and also brought up by reviewer 1, we have revisited the manuscript and improved the description of how extremes are defined across sub-projects. However, as mentioned by the reviewer, the sections have different threshold and associated definitions of extremes and we tried to explain them as much as possible. We agree that the different extremes have different representations of the warm seasons. We did not want to harmonize the analysis choosing the same definition of summer, rather focusing on the particularity of the extreme and compound for the months where it was most expressed. We tried to be more specific in some places about the choice of seasonality.

*150: Note that using a 15 day running window do define extreme thresholds (as done Fischer and Schär) of has been shown to be methodologically wrong and prone to biases recently:* [https://www.nature.com/articles/s41467-024-46349-x](https://www.nature.com/articles/s41467-024-46349-x)

Thank you very much for making us aware of this work, which was published well after we submitted our manuscript. However, the paper is referring to biases due to long running windows of 31 days, while at the same time draws attention to the bias of very short running windows of 5 days due to the very small samples. We are thus not changing the selected 15-day running window that does not fall to either risk.

*229: "The co-occurrence should be on the same day or the following day for precipitation, in the same grid cell and within a radius of 50 km, respectively." I'm not sure I understand what it means for an event to be within the same grid cell and at the same time within 50km. If we assume the 0.25deg resolution corresponds to 25km, this would mean it can be a maximum of two grid cells away in the horizontal as an example? Is that correct? Can the authors clarify this?*

240: We have rewritten the section:
"Co-occurrence is defined when wind gusts and precipitation both exceed their respective 98th percentile at a specific grid box, with precipitation exceedance occurring on the same day, the day before, or the day after, within a 50 km radius around the grid box center."

*274: "The compound precipitation-wind events are defined on the winter (December to February) daily mean precipitation and daily maximum surface wind. The selection of events is based on the exceedance of the 98th percentile for the period 1975-2025…"*

*and then later*

*278: "Extreme compound wind and precipitation years exceed the 20-year return levels for precipitation and wind individually, defined as the 95th percentiles for the period 1975-2025."*

*Sorry but this confuses me. Can the authors clarify the difference between "compound precipitation-wind events" and " Extreme compound wind and precipitation years"?*

In both cases, we are referring to years, as compound events are defined as multi-month or seasonal occurrences of extreme events. We have further rewritten the passage defining the thresholds:

301: "The selection of events is based on the exceedance of the 98th percentile for wind and precipitation during the period from 1975 to 2025. For each grid cell, wind and precipitation events are identified when they exceed this threshold on the same day, while for precipitation alone, the exceedance can occur either on the same day or the day after."

*303: "a large blocking system at 500 hPa, and a double jet stream configuration is visible in the 250mb zonal wind field"*

*Maybe settle for one unit?*

We rephrased the section using the same unit, here, and throughout the whole manuscript:

329: "a large blocking system at 500 hPa and a double jet stream configuration in the 250 hPa zonal wind field"

*Also I read the sentence as saying that a 500hPa blocking is visible in the 250mb zonal wind field. I think I understand what the authors try to say here but you can not see 500hPa at 250hPa by definition…*

330: The blocking at 500 hPa information stems from the paper of Roussi et al. (2023), which we have wrongly cited a bit later in the sentence. We now corrected it In our figure we present only 250 hPa zonal wind and anomalies.

*Figure 12: what do the dashed lines represent?*

The dashed lines denote the climate variable thresholds that define the extreme CEs. It is now better clarified in the text.

*Figure 13: Fontsize way too small*

We adjusted the font size and made the figure more accessible to the reader.

*(b) the title says CMIP5 1975-2021, the caption says historical CMIP5 ensemble. Maybe I'm not aware that the CMIP5 historical runs where extended otherwise this is wrong.*

The historical CMIP5 ensemble used in this figure has been extended by Aalbers et al. (2023). We have revised the text and the figure caption as well.
312: … and for bilinearly interpolated (regular 0.5° lon-lat grid) and extended to 2021 ensemble simulations of CMIP5 (Taylor et al., 2012; Aalbers et al., 2023)…

*"Significant model ensemble grid points are shaded dark." Unclear what that dark means here.*

619ff: The caption and description of figure 13 have been rewritten to better clarify the meaning of dark green and dark brown.

*614: "The showcases presented in this paper include multivariate, pre-conditioning, temporally and spatial Ces"*

*Again, a summarizing table would be nice to give an overview.*

We agree that a table would enhance the clarity and provide a concise overview of the key aspects discussed. However, in order not to extend the paper even more, we would like not to add this table here. Instead, we tried to make the different aspects mentioned by the reviewer more clearly.

*611-648: Delete or shorten by a lot? This is not a conclusion but basically a reiteration of the results.*

643ff: We have rewritten the last section completely, our conclusions are the findings of the case studies, identifying the lack of accurate definitions of compound events beyond of single case studies, requirement of better understanding of drivers and factors that lead to those events and beyond to their impacts.

*655: "Further studies aim at expanding the current knowledge on the complex relationships between CEs and large-scale fields at different time horizons in order to improve the detection and thus the understanding of the climate system."*

*Not sure what this sentence is trying to say? What further studies?*

643ff: We rewrote the last section of the manuscript completely.

In addition, we updated the literature with new publications that we think are important to cite in the different sections

**Editor comments**

*Line 74/75: "specifically FOR adaptation planning"*

We have made the change, thank you.

*Line 83: the statement is difficult to read and it is not exactly clear, what kind of "processes" are meant. Perhaps just saying "… it is essential to adapt research strategies and tools (e.g. models) to incorporate …"*
*The passage has been rewritten:*
77: Current research on weather and climate impacts, risks and damages often underestimates the influence of CEs (Ridder et al., 2021). It is therefore essential to adapt research strategies and tools, such as models, to integrate compound weather and climate events , enabling a more accurate assessment of uncertainties, impacts and risks.

*Line 86: "drivers" instead of "drives"?*
We have made the change.

*Line 89: "confident" instead of "confidence"?*
As the assessments characterisations are mainly provided with confidence levels rather than adjectives, we have thus not followed the suggestion.

*Line 103: "answers to questions" or "answer questions"?*
This passage is no longer part of the manuscript following the reviewers' comments.

*Line 120: I don't see these numbers for Germany in either Toreti 2019a or 2019 b. Please check the reference. These numbers can easily be derived from DWD-data (e.g. Klimastatusreport 2018).*
112: We have updated the numbers and the source.

*Line 129: Is it technically correct to say that the rivers could not cool the reactors, or is that due to regulations to avoid too high temperatures in the rivers? Perhaps it could be rephrased as "… the rivers could not provide sufficient cooling capacity for the reactors"?*
We have revised the sentence accordingly, thank you.

*Line 164: "hydrological year 2017/2018"?*
Thank you for this comment. We have looked once again in detail within Hartick et al. (2021), where they explicitly define the hydrological year 2018/2019.

*Line 240: I am not sure if I understand that statement: Does that mean that every (significant) trend is removed?*
The section on 2.8 Concurrent heavy rain and storm extremes – estimation of probability of event occurrence has been rewritten and the statement clarified.

*Line 305: "hot summer of 2003"*
We have made the change.

*Line 368: "German national institutes": could you be more specific or provide a reference? Is it e.g. https://www.dwd.de/bodenfeuchteviewer ?*
397: we have introduced the Destatis

*Line 590: "reanalysis" instead of "Reanalysis",*
We have made the change.